# UniHG: A Large-scale Universal Heterogeneous Graph Dataset and Benchmark for Representation Learning and Cross-Domain Transferring

**Yide Qiu[1], Tong Zhang[1,*], Shaoxiang Ling[1], Xing Cai[1], Ziqi Gu[1], Zhen Cui[2,*]**

[1]School of Computer Science and Engineering, Nanjing University of Science and Technology
[2]School of Artificial Intelligence, Beijing Normal University
{q115025886, tong.zhang, lingshaoxiang}@njust.edu.cn;
caixing11@163.com; ziqigu@njust.edu.cn; zhen.cui@bnu.edu.cn

## Abstract

Irregular data in the real world are usually organized as heterogeneous graphs consisting of multiple types of nodes and edges. However, current heterogeneous graph research confronts three fundamental challenges: i) Benchmark Deficiency, ii) Semantic Disalignment, and iii) Propagation Degradation. In this paper, we construct a large-scale, universal, and joint multi-domain heterogeneous graph dataset named **UniHG** to facilitate heterogeneous graph representation learning and cross-domain knowledge mining. Overall, UniHG contains 77.31 million nodes and 564 million directed edges with thousands of labels and attributes, which is currently the largest **universal** heterogeneous graph dataset available to the best of our knowledge. To perform effective learning and provide comprehensively benchmarks on UniHG , two key measures are taken, including i) the semantic alignment strategy for multi-attribute entities, which projects the feature description of multi-attribute nodes and edges into a common embedding space to facilitate information aggregation; ii) proposing the novel Heterogeneous Graph Decoupling (HGD) framework with a specifically designed Anisotropy Feature Propagation (AFP) module for learning effective multi-hop anisotropic propagation kernels. These two strategies enable efficient information propagation among a tremendous number of multi-attribute entities and meanwhile mine multi-attribute association adaptively through the multi-hop aggregation in large-scale heterogeneous graphs. Comprehensive benchmark results demonstrate that our model significantly outperforms existing methods with an accuracy improvement of 28.93%. And the UniHG can facilitate downstream tasks, achieving an NDCG@20 improvement rate of 11.48% and 11.71%. The UniHG dataset and benchmark codes have been released at `https://github.com/Yide-Qiu/UniHG`.

## 1 Introduction

Heterogeneous Graphs (HGs), also known as Heterogeneous Information Networks (HINs), consist of multiple types of nodes and links. Compared to homogeneous graphs, the heterogeneity in multi-attribute nodes and graph topology makes HGs carry richer semantics and be more suitable to characterize a variety of complex real-world systems such as academic networks and social networks. For this reason, methods that focus on representation learning on HGs have drawn increasing attention in recent years, and have been applied to tackle numerous tasks in diverse domains, including recommendation systems Gao et al. (2022); Wu et al. (2022), malware detection systems Zhang et al. (2022a), and healthcare systems Cao et al. (2022).

---

*Corresponding author: Tong Zhang and Zhen Cui.

39th Conference on Neural Information Processing Systems (NeurIPS 2025) Track on Datasets and Benchmarks.

Table 1: Statistics for the proposed UniHG and other heterogeneous graph datasets.

| Datasets | #Nodes | #Node Types | #Edges | #Edge Types | #Task | #Labels | #Domain |
|---|---|---|---|---|---|---|---|
| ACM | 10,942 | 4 | 547,872 | 8 | Node Level | 3 | Citation |
| IMDB | 21,420 | 4 | 86,642 | 6 | Node Level | 5 | Recommendation |
| DBLP | 26,128 | 4 | 239,566 | 6 | Node Level | 4 | Citation |
| LastFM | 20,612 | 3 | 141,521 | 3 | Edge Level | 2 | Social Network |
| PubMed | 63,109 | 4 | 244,986 | 10 | Edge Level | 2 | Medicine |
| Freebase-book | 180,098 | **8** | 1,057,688 | 36 | Node Level | 7 | Recommendation |
| Amazon Review | 102,699,417 | 2 | 571,544,897 | 2 | Node Level | 33 | Recommendation |
| MAG240M | **244,160,499** | 3 | **1,728,364,232** | 3 | Node Level | 153 | Citation |
| UniHG (ours) | 77,312,474 | 1 | 564,425,621 | **2,082** | Node Level | **74,666** | **Universal** |

Existing HG-related works involve both the construction of HG datasets and the design of effective learning methods, where an encyclopedic HG dataset may be especially crucial to promote HG learning. Regarding HG dataset construction, previous studies Tang & Liu (2009); Ma et al. (2011); Sun et al. (2011a); Cantador et al. (2010); Sen et al. (2008) focus on constructing small-scale or domain-specific HG datasets. For instance, IMDB, which contains 10,942 nodes, is a small-scale recommendation system dataset. While MAG240M, despite containing more than 244 million nodes, only encompasses three types of nodes and edges. For the learning methods on HGs, existing approaches can be categorized into aggregation-based methods Hu et al. (2020b); Veličković et al. (2017); Hong et al. (2020); Yang et al. (2021) and meta-path-based methods Wang et al. (2020); Huang et al. (2016); Sun et al. (2011b); Chang et al. (2022), depending on how they capture semantic structure. Aggregation-based methods primarily focus on iteratively aggregating multi-hop neighbor information from sampled subgraphs to understand heterogeneous topological characteristics. Moreover, meta-path-based methods aim to explore representative patterns from multiple sampled meta-paths to learn heterogeneous topological representations. For more detailed expositions on related work, please refer to Appendix B.

Although existing works on HGs have achieved notable successes, there are still several challenges to be addressed. One critical issue is the lack of a large-scale universal HG dataset that can express comprehensive real-world knowledge. For existing HG datasets, the types of entities and relationships are rather limited, which greatly limits their capacity to facilitate heterogeneous graph representation and abundant real-world knowledge extraction. Additionally, there remains a deficiency of methods that are both effective and efficient on large-scale universal HG. The process of sampling and aggregating subgraphs of existing aggregation-based methods usually leads to the loss of topology information or the introduction of additional processing times when dealing with large-scale HGs with numerous relations. Similarly, as the number of association types increases, the quantity of meta-paths can grow exponentially, which results in these methods facing challenges in accurately perceiving heterogeneous topology. Therefore, there is an urgent need for a large-scale universal HG dataset that encompasses sufficient types of real-world entities and relations, along with comprehensive effective benchmarks.

To address the aforementioned challenges and provide the comprehensive benchmark, we propose a novel universal HG dataset, UniHG, along with a corresponding learning method called Heterogeneous Graph Decoupling (HGD). The overall pipeline is illustrated in Figure 5. The proposed dataset is constructed by extracting a vast number of multi-attribute entities and relations from Wikidata contributors (2024). Specifically, UniHG comprises more than 77.31 million nodes and 564 million directed edges with 2082 diverse association types. To enable effective learning on such a large-scale HG dataset, two key strategies are employed. i) we introduce a semantic alignment strategy, projecting the feature descriptions of multi-attribute entities and relations from Wikidata into a common HG embedding space. ii) we propose a novel HGD framework with a specially designed Anisotropic Feature Propagation (AFP) module for learning effective multi-hop anisotropic propagation kernels. These strategies enable efficient representation learning and comprehensive benchmark on large-scale universal HGs. We conduct extensive benchmark on UniHG using HGD and eight existing baseline methods. The experimental results demonstrate that our proposed HGD achieves state-of-the-art performance and is approximately 22.1 times faster than existing methods. Furthermore, we transfer the universal knowledge learned from UniHG to recommendation tasks. The corresponding experiments validate the effectiveness of generalizing to graph downstream tasks.

Our contributions can be summarized as follows:

1) We construct a large-scale universal heterogeneous graph dataset, UniHG, to support HG learning. Compared to existing HG, UniHG provides an advantage in universal knowledge representation.

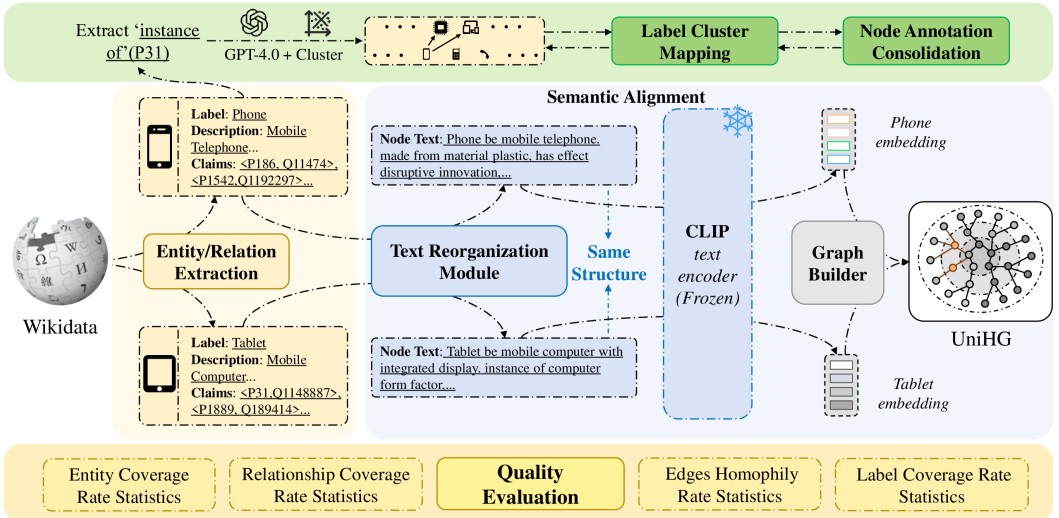

Figure 1: **Overall architecture** of UniHG construction. We describe this process by the example of phones and tablets. First, all structured information is fed into a text reorganization module. It is organized into paragraphs with a consistent structure. Then, the effective attributes of entities are processed through a pre-trained language model to derive their node representations. On the top of the figure, the 'instance of' relation is used to generate node labels via a label cluster mapping module. Other relations are transformed into directed edges by the graph builder. In the quality evaluation at the bottom of the figure, we record four metrics to evaluate UniHG.

2) We propose the novel Heterogeneous Graph Decoupling (HGD) framework, featuring a specially designed Anisotropic Feature Propagation (AFP) Module for efficient learning on large-scale HGs.

3) Comprehensive benchmark validates the effectiveness of the proposed UniHG and HGD models, as well as the transferability of universal knowledge from UniHG.

## 2 UniHG Construction

### 2.1 Overview

In this section, we introduce the construction of the large-scale universal graph dataset, Universal Heterogeneous Graph (**UniHG**), including the construction process, data statistics, and dataset evaluations. We adhere to the principle of semantic maximization and follow an automated pipeline of 'Entity/Relation Extraction and Filtering - Semantic Alignment - Node Annotation'. Firstly, compare to Wikidata, the key differences are summarized as follows: i) **Structural Differences**: Publicly available Wikidata consists of entities and relationships represented by structured text, rather than an explicit graph structure. In contrast, UniHG is a large-scale, universal heterogeneous graph dataset, constructed through comprehensive processing and manual refinement. ii) **Application Scenario Differences**: Wikidata is primarily utilized for knowledge retrieval tasks and is not designed for graph learning. Conversely, UniHG is specifically tailored for graph neural network research, supporting learning on large-scale heterogeneous graphs. iii) **Storage Overhead Differences**: Wikidata, comprising structured text, results in a complete JSON storage file exceeding **1.7TB**. This massive size limits its wide applicability for deep learning. In contrast, UniHG is constructed using low-dimensional dense node feature tensors. The complete feature tensor matrix requires only **37.52GB**, significantly reducing the storage overhead and making it more suitable for broad use. In summary, UniHG is a large-scale, cross-domain, multi-attribute universal heterogeneous graph containing **77,312,474** nodes and **564,425,621** directed edges of **2,082** different types. Please see Figure 1 and Table 6 for the overall architecture and statistics of UniHG construction.

### 2.2 Entity/Relation Extraction

Extracting semantically enriched entities and relationships from the vast web-scale knowledge graph is a critical step in UniHG construction. However, mining the valuable attributes of each entity from billions of intricately structured data entries in Wikidata presents a formidable challenge.

> **Format of Entities**: Feature node. ID: <Content>; Label: <Label content>; Description: <Description content>.
> **Example**: Feature node. ID: Q31; Label: Belgium; Description: Country in western Europe.
> **Example**: Feature node. ID: Q34660; Label: J. K. Rowling; Description: British author and philanthropist (born 1965).

To address this hurdle, we design a standardized extraction format to capture textual depiction of each entity and relation, which aims to excavate and synthesize textual descriptions capable of comprehensively portraying entities with semantic attributes. Furthermore, Wikidata hosts a multitude of semantically inconsequential links, such as 'external-ids', which need to filtrate. While other associations deemed semantic will be integrated into UniHG as typed directed edges. The formats for entity and relationship extraction and illustrative examples are presented.

> **Format of Relations**: Feature edge. Triplet: <Content>; Label: <Label content>; Description: <Description content>.
> **Example**: Feature edge. Triplet: <Q31, P1344, Q1088364>; Label: Participant in; Description: Q31 Participant in Q1088364.
> **Example**: Feature edge. Triplet: <Q34660, P800, Q8337>; Label: notable work; Description: Q34660 notable work Q8337.

## 2.3 Semantic Alignment

Consistent text structure and common representation space benefit language model understanding and training. Thus we design a text structure reorganization-based alignment strategy, so that entities with different text structures and associations can be embed into a common representation space. Considering the fragmentation and disordering of information extraction, the textual features of nodes are described as a ordered sentence with a specific structure '<entity info>, <relationship1 info>, <relationship2 info>, ... '. The '<entity info>' is represented as 'Label be Description' and <relationship info>denotes the corresponding relational Description. In this way, each entity is automatically organized into a consistent structure of feature descriptions. We depict each node using entity description and related relationship descriptions rather than relying solely on single words. This approach aggregates local semantics at the text level and mitigates the problems of synonymy and polysemy. Then, the CLIP Radford et al. (2021) text encoder is used to projects the feature descriptions of all entities and relations into a common representation space. Through consistent textual structure and text embedding extractor, we align the feature space of each nodes and edges.

> **Text Structure of Nodes**: <Entity info>, <Relation1 info>, <Relation2 info>, ...
> **Example**: Belgium be Country in western Europe. Belgium Participant in Battle of The Lys...
> **Example**: J. K. Rowling be British author and philanthropist (born 1965). J. K. Rowling notable work Harry Potter...

## 2.4 Node Annotation Consolidation

The dataset requires precise annotation and verification. Firstly, we observe that Wikidata utilizes the 'instance of (P31)' property to represent entity annotations. However, these annotations are often overly redundant, such as 'fictional character', 'conceptual character', 'fictional character in a musical work', and 'imaginary character'. Such synonym redundancy may confuse the distribution of label semantics with feature semantics, potentially misleading model training. Furthermore, there are a total of 74,666 annotations in Wikidata, leading to exceedingly long predicted probability vectors, which introduces significant computational overhead and hinders model convergence. To construct the training labels for UniHG, we adopt a strategy combining a 'self-supervised clustering algorithm and a large language model', and invest significant human resources for annotation checks. Specifically, we design a label clustering and mapping module to consolidate highly similar categories. The process is as follows: i) (extract text representation) we use the text encoder from CLIP Radford et al. (2021) to extract the text representations of the original labels. ii) (representation cluster) these representations are then clustered in a self-supervised manner into several label clusters. iii)

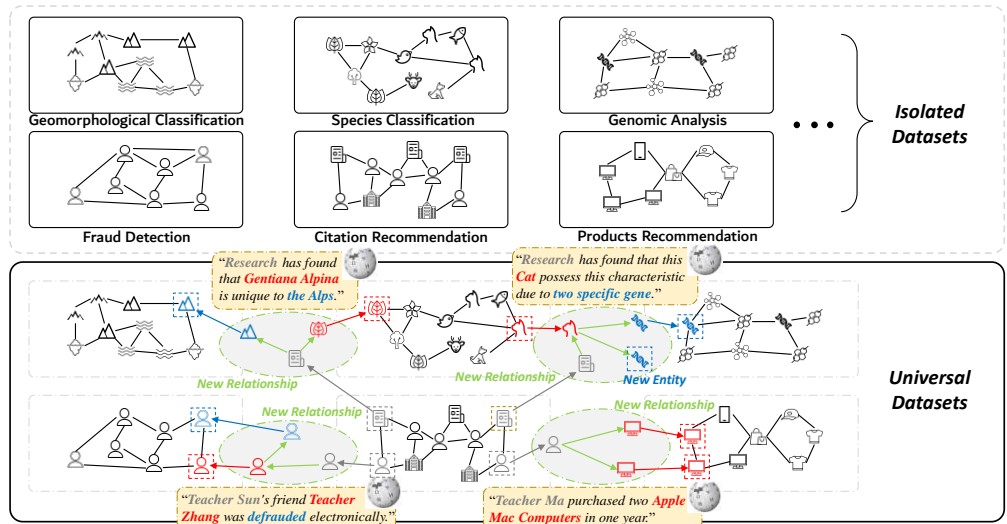

Figure 2: Comparison between **isolated datasets** and **universal datasets**. For example, according to the '**Research** has found that this **Cat** possess this characteristic due to **two specific gene**.' in Wikidata, we can construct a new universal relationship among a **citation** in the Citation Recommendation dataset, a **cat** in the Species Classification dataset, and a **gene fragment** (as well as a **gene fragment** outside of the dataset) in the Genomic Analysis dataset.

(text-to-text generation) the textual descriptions of the label clusters are generated using GPT-4.0 based on the text of the original labels. Subsequently, we manually verify the mapping relationships between the original and cluster labels. In this process, 37 researchers are hired, each reviewing 150 mapping relationships at a rate of $1 per relationship. Each mapping is checked by at least three researchers. A mapping is consider correct if all manual checks are positive; otherwise, it is flagged for further review and correction by additional researchers. This process follows a standard dataset construction protocol Ratner et al. (2016), and the number of cluster can be further adjusted based on manual verification. After multiple iterations of the label clustering and mapping module, the 74,666 labels are consolidated into 2,000 cluster labels, serving as the supervisory signal for representation learning. As a result, each node is annotated with two vectors (one is the original multi-label vector with size $\mathbb{R}^{1\times74666}$ and another is the training multi-label vector with size $\mathbb{R}^{1\times2000}$), and each edge is annotated with a vector of size $\mathbb{R}^{1\times2082}$, where 2,082 corresponds to the number of relation types in Wikidata. Additional annotation details, including original labels, mapping examples, verification logs, and GPT prompts, are provided in the Anonymous GitHub link.

## 2.5 Quality Evaluation

To evaluate the quality of UniHG, we conduct a comprehensive evaluation of the following aspects: i) Category distribution of entities in UniHG. ii) Word cloud visualization of the original label texts. The top three frequently occurring words are 'compounds', 'occurrence', and 'genre'. iii) Numerical statistics of UniHG and its two subsets. iv) Four dataset statistical metrics. The Entities Coverage Rate is quantified at 0.73, indicating that UniHG includes most of the entities in Wikidata. The Relations Coverage Rate is 0.19, due to 63% of the relations are external-ids types. And the Labels Coverage Rate stands at 0.87, indicating that the scope of knowledge approximates the Wikidata. Furthermore, the Edges Homophily Rate is computed to be 0.58. Please see Figure 3 for these statistics.

## 2.6 Visualization of Comparison with Isolated Datasets

A comparative analysis between UniHG and existing domain-specific datasets (referred to as isolated datasets). Figure 2 illustrates that, in contrast to existing isolated datasets, UniHG leverages the multi-attribute associations of cross-domain entities sourced from Wikidata. These associations serve as bridges connecting disparate isolated datasets, thereby facilitating the establishment of meaningful relationships between entities spanning diverse domains. This integration of rich cross-domain associations transcends the constraints of existing isolated datasets, thereby offering significant support for the future research.

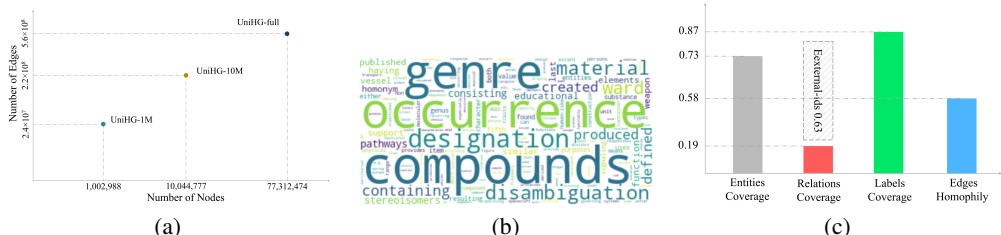

|     |     |     |
| --- | --- | --- |
| (a) | (b) | (c) |

Figure 3: (a) The statistics of UniHG, (b) the word cloud figure of label texts, and (c) the four statistical indicators of UniHG-full.

## 2.7 Dataset Update

Frequent whole graph updates to UniHG may require high overhead. To update the text embeddings, we design the following update strategy: i) Adding a New Node/Edge: The UniHG construction process allows direct updates using the extraction module and the frozen language model. For pre-trained node embeddings, only embeddings within a 2-hop neighborhood of the new node/edge need to be updated, which reduces the update cost. ii) Updating a Large Number of Nodes or Edges: A global update of UniHG is performed when necessary. On a single RTX 4090 GPU, a global update takes approximately 70 hours.

## 3 Heterogeneous Graph Decoupling

To provide comprehensive benchmark on large-scale universal heterogeneous graphs, we further propose a novel representation learning framework, Heterogeneous Graph Decoupling (HGD), for effectively learning the multi-attribute representation of constructed UniHG. This framework primarily consists of two processes: train-free anisotropic feature propagation and graph-free feature mapping. Specifically, the anisotropic feature propagation is designed to embed multi-attribute structural information into multi-hop propagation features. This enables the feature mapping learning of information-rich multi-hop propagation features to approximate full graph convolution in the neighborhood direction. Please see Appendix A for the complete notation.

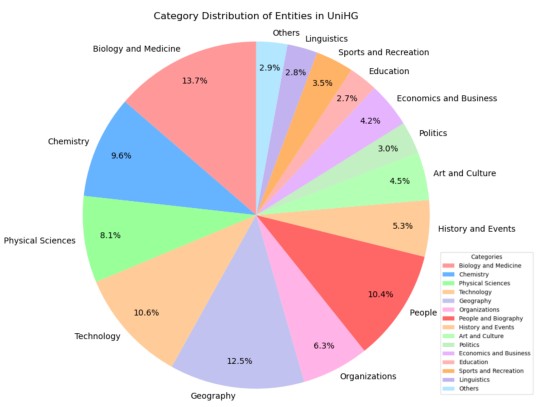

Figure 4: Distribution of node categories of UniHG.

**Anisotropic Feature Propagation** We firstly present the definition of one-hop anisotropic feature propagation: propagating one-hop messages in advance along distinct heterogeneous edges in CPU without trainable parameters, which can be extended to multi-hop propagation accordingly.

Specifically, for a given heterogeneous graph $\mathcal{G} = (\mathcal{V}, \mathcal{E})$, where $\mathcal{V}$ and $\mathcal{E}$ represent the sets of nodes and edges, respectively. Each node $v_i \in \mathcal{V}$ and each edge $e_i \in \mathcal{E}$ is associated with only one specific type. Assuming that heterogeneous graph's node and edge information can be mapped into the common space, the adjacency matrix $\mathbf{A} \in \mathbb{R}^{N \times N}$ can be represented as a relation-aware adjacency matrix $\hat{\mathbf{A}} \in \mathbb{R}^{N \times N \times d}$. Therefore, the paradigm of anisotropic feature propagation can be formulated as:

$$\mathbf{Z} = \delta([\mathbf{X}|\mathbf{C}^1|\mathbf{C}^2|\dots|\mathbf{C}^K]), \tag{1}$$

where $K$ is the number of anisotropic feature propagation steps. For each hop $k \in [1, K]$, $\mathbf{C}^k \in \mathbb{R}^{N \times d}$ denotes the anisotropic propagation matrix of the $k$-th hop. The function $\delta$ encompasses both learning and non-learning aggregation schemes for multi-hop features. The propagation of each hop can be formulated as an iterative function $f(\cdot)$. Thus, we have $\mathbf{C}^{k+1} = f(\hat{\mathbf{A}}, \mathbf{C}^k)$ and $\mathbf{C}^0 = \mathbf{X}$, where $\mathbf{X}$ is the input node feature matrix. To accelerate this process in parallel on large-scale graphs, we first upgrade the dimension of each $\mathbf{C}$ as follows: $\hat{\mathbf{C}}i, j, \cdot = \mathbf{C}i, \cdot$ for all $i, j \in 1, 2, \dots, n$. Then

Table 2: Results of comparison experiments on UniHG-1M, UniHG-10M, and UniHG-full. Our proposed method HGD and the best performances are highlighted in **bold**.

| Methods | Param. | UniHG-1M | | | UniHG-10M | | | UniHG-full | | |
|---------|--------|----------|--------|----------|-----------|--------|----------|------------|--------|----------|
| | | Accuracy | Recall | F1 score | Accuracy | Recall | F1 score | Accuracy | Recall | F1 score |
| GCN Chen et al. (2018) | 1.48M | 23.68 | 25.26 | 21.89 | 22.86 | 25.66 | 23.19 | 26.72 | 24.19 | 23.26 |
| GAT Veličković et al. (2017) | 1.49M | 29.12 | 34.02 | 27.76 | 33.73 | 37.48 | 29.88 | 31.02 | 35.07 | 26.92 |
| HGT Hu et al. (2020b) | 411M | 51.28 | 55.30 | 51.55 | 52.09 | 56.52 | 53.03 | 55.36 | 61.91 | 56.47 |
| MTMP Pei et al. (2024) | 2.36M | 47.52 | 47.48 | 60.79 | 59.95 | 61.33 | 72.81 | 65.67 | 66.15 | 67.74 |
| SGC Wu et al. (2019) | 0.61M | 42.56 | 42.11 | 55.12 | 56.32 | 57.78 | 67.95 | 62.15 | 63.88 | 73.21 |
| SIGN Frasca et al. (2020) | 1.73M | 56.73 | 55.54 | 69.41 | 73.58 | 84.17 | 80.30 | 69.04 | 70.49 | 81.48 |
| GAMLP Zhang et al. (2022c) | 0.64M | 44.55 | 43.74 | 56.92 | 59.47 | 61.32 | 70.02 | 64.23 | 66.05 | 74.34 |
| **HGD (ours)** | 1.51M | **75.41** | **75.95** | **82.64** | **89.03** | **90.11** | **93.05** | **93.16** | **93.83** | **96.09** |

we propose to utilize the Dimension Adaptive Tensor Product Gerstner & Griebel (2003) to compute $f(\hat{\mathbf{A}}, \hat{\mathbf{C}}^k)$, which can be formulated as:

$$\mathbf{H}^{k+1} = f(\hat{\mathbf{A}}, \hat{\mathbf{C}}^k) = \hat{\mathbf{A}} \times_2 \hat{\mathbf{C}}^k, \tag{2}$$

where $\times_2$ denotes the adaptive tensor product along the second dimension. The $\mathbf{H}^{k+1}$ represents the message passing matrix along edge features for each node. Considering the adaptive information aggregation for in-degree messages, we apply a row-wise softmax function along the second dimension of $\mathbf{H}^{k+1}$, followed by an $l_2$-norm on the vectors in the third dimension:

$$\mathbf{C}^{k+1} = Softmax(||\mathbf{H}^{k+1}||_2). \tag{3}$$

Finally, the $k$-hop representation can be obtained through $\mathbf{Z}^k = \zeta^k(\mathbf{C}^k)$, where the $\zeta$ represents a MLP encoder. AFP module's convergence upper bound can be proved as:

$$||\mathbf{C}^k - \mathbf{C}^*||_F = \frac{L^k}{1-L}||\mathbf{C}^0 - \mathbf{C}^1||_F, \tag{4}$$

where $\mathbf{C}^*$ is a unique fixed point. Please see the C.1 for detailed derivation.

**Graph-free Feature Mapping**   The feature mapping process aims to learn effective representations from multi-hop propagation features. Considering each node may pay different attention to multi-hop features, we introduce a hop-wise attention mechanism. This mechanism computes the weighted sum of multi-hop representations with the diagonal attention matrix Sun et al. (2020) to adaptively aggregate multi-hop information, which reduces the dependence on graph topology information and can be approximated as graph-free mapping. This process can be formalized as:

$$\mathbf{Z}_{out} = \xi(\sum_{k=0}^{K} \mathbf{\Theta}^k \mathbf{Z}^k + \mathbf{X}\mathbf{W}_r), \tag{5}$$

where $\xi$ denotes an encoder with a step connection matrix $\mathbf{W}_r$, $\mathbf{\Theta}^k$ represents the $k$-th diagonal attention matrix. To effectively learn the probability distribution of nodes across multi-labels, the widely used Binary Cross-Entropy (BCE) loss is employed as the learning objective:

$$\mathcal{L} = -\frac{1}{N}\sum_{i=1}^{N}\left[\mathbf{y}_i \cdot \log(\hat{\mathbf{y}}_i) + (1 - \mathbf{y}_i) \cdot \log(1 - \hat{\mathbf{y}}_i)\right], \tag{6}$$

where $N$ is the number of nodes, $\mathbf{y}_i$ and $\hat{\mathbf{y}}_i$ denote the ground truth label vector and the predicted label vector of node $i$. Please see the appendix C.3 for the spatiotemporal complexity analysis.

## 4   Experiments

The experimental section evaluates the potential of HGD framework and the transfer effectiveness of UniHG dataset by answering the following questions: **Q1**: How does the proposed HGD effectively learn node representations on such a large-scale heterogeneous graph? **Q2**: How does anisotropic feature propagation affect GNN performance? **Q3**: How much are the spatial cost and temporal cost of the decoupling framework on UniHG? **Q4**: How does the universal knowledge in UniHG facilitate downstream graph tasks? By answering these questions, we validate the capability of HGD in addressing large-scale heterogeneous graph representation learning tasks and the significant potential of transferring universal knowledge of UniHG to enhance various downstream graph tasks. Please see Appendix D for more experiments and training details.

Table 3: **Results of the ablation experiments of AFP module** on UniHG-1M, UniHG-10M, and UniHG-full. '-AFP' means 'using the pre-calculated feature of AFP module' and the better performances are highlighted in **bold**.

| Methods | UniHG-1M | | | UniHG-10M | | | UniHG-Full | | |
|---|---|---|---|---|---|---|---|---|---|
| | Accuracy | Recall | F1 score | Accuracy | Recall | F1 score | Accuracy | Recall | F1 score |
| SGC | 42.56 | 42.11 | 55.12 | 56.32 | 57.78 | 67.95 | 62.15 | 63.88 | 73.21 |
| SGC-AFP (ours) | **44.18** | **42.75** | **57.65** | **64.25** | **65.21** | **76.67** | **69.84** | **71.40** | **81.45** |
| SIGN | 56.73 | 55.54 | 69.41 | 73.58 | 84.17 | 80.30 | 69.04 | 70.49 | 81.48 |
| SIGN-AFP (ours) | **66.69** | **65.27** | **77.16** | **77.52** | 80.18 | **81.38** | **75.45** | **76.42** | **85.71** |
| GAMLP | 44.55 | 43.74 | 56.92 | 59.47 | 61.32 | 70.02 | 64.23 | 66.05 | 74.34 |
| GAMLP-AFP (ours) | **47.24** | **46.80** | **60.85** | **60.99** | **62.70** | **73.38** | **72.89** | **74.95** | **83.51** |

**Datasets** Overall, we utilized six datasets, including UniHG-1M, UniHG-10M, and UniHG-full for multi-label node classification tasks, and Amazon Book, Yelp2018, and Citeulike-a for downstream recommendation tasks (knowledge transfer experiments). UniHG-1M and UniHG-10M are subsets of UniHG. These subsets are created by applying Snowball Sampling Goodman (1960) to sample over 1 million nodes or 10 million nodes from UniHG. Table 6 shows the statistics of the three scales of UniHG. Each UniHG dataset is randomly split into training, validation, and test sets with a ratio of 8:1:1.

**Evaluation Metrics** For the multi-label node classification task, we employ the popular metrics such as 'Accuracy', 'Recall', and 'F1 Score'. Specifically, we use 'Subset Accuracy' as 'Accuracy' metric, which measures the proportion of nodes for which the predicted labels exactly match the ground truth labels. For the recommendation task, we utilize widely-used metrics including 'Precision@20', 'Recall@20', and 'NDCG@20' (Normalized Discounted Cumulative Gain).

## 4.1 Node Classification Tasks for UniHG

**Experiment Setting** To answer **Q1**-**Q3**, we conduct experiments using HGD and various baseline methods on the semi-supervised multi-label node classification task on UniHG. Specifically, we evaluate three different scales of UniHG: UniHG-1M, UniHG-10M, and UniHG-full. UniHG-1M and UniHG-10M are subsets of UniHG, constructed by applying Snowball Sampling Goodman (1960) to sample over 1 million and 10 million nodes, respectively. Table 6 provides the statistics for these three scales of UniHG. For baseline methods, we employ three convolution-based methods: **GCN**Chen et al. (2018), **GAT**Veličković et al. (2017), and **HGT**Hu et al. (2020b), as well as three decoupling-based methods: **SGC**Wu et al. (2019), **SIGN**Frasca et al. (2020), and **GAMLP**Zhang et al. (2022c). Please see Appendix D.2 for the discussion of other two baselines.

**Experiment Analysis** From the results shown in Tables 2, Tables 3 and Figure 7 in Appendix, we have the following observations: i) (for **Q1**) The proposed HGD outperforms all baseline methods on all three scales of UniHG, indicating that HGD can effectively learn representations from large-scale heterogeneous graphs, which are typically challenging for GNNs. ii) (for **Q2**) AFP successfully enables decoupled methods to perform effectively on all scales of UniHG. All backbone models with the anisotropic feature propagation module perform well on UniHG. This suggests that anisotropic decoupling allows multi-attribute relational features to be well-aggregated into node features. Methods that can learn multi-hop features more thoroughly may have the potential to further improve performance. We believe the effectiveness of AFP requires further investigation and leave it to our future works. iii) (for **Q3**) The HGD framework achieves more better balance between computational efficiency and model performance. Although GCN and GAT infer faster due to fewer parameters, their performance is poorer across all dataset scales. This may be due to feature distribution biases on both each relation subgraphs and batch subgraphs. HGT incurs significant computational overhead, which may be due to the storage and learning of multi-head attention parameter matrices for each relation, which hinders its learning on UniHG. In contrast, the fast approximate tensor product implementation in AFP exhibits few overheads. Graph-free feature mapping is simpler and faster than multi-head attention learning for each relation, making the inference speed of HGD over 22.1 times faster than HGT. For the current experiments, we have compared classic and effective methods, such as traditional graph convolution methods (GCN and GAT) and graph decoupling methods (SGC, SIGN, and GAMLP). Due to the classical nature of computational theory, these methods can serve as representatives of the series of network architectures. Please see Appendix D.2 for discussions on other methods (ieHGCN Yang et al. (2021), SeHGNN Yang et al. (2023) and HINormer Mao et al. (2023)), where ieHGCN and SeHGNN represent a series of meta-path based methods.

Table 4: Results of knowledge transfer experiments on recommendation system. The 'Method$_{\text{-UniHG}}$' means performance improvement ratio of using universal knowledge from UniHG-full on Amazon-book, Yelp2018, and Citeulike-a.

| Methods | Amazon-book | | | Yelp2018 | | | Citeulike-a | | |
|---|---|---|---|---|---|---|---|---|---|
| | Precision@20 | Recall@20 | NDCG@20 | Precision@20 | Recall@20 | NDCG@20 | Precision@20 | Recall@20 | NDCG@20 |
| LightGCN | 0.01716 | 0.06191 | 0.04106 | 0.00433 | 0.01123 | 0.00849 | 0.02329 | 0.07188 | 0.05064 |
| LightGCN$_{\text{-UniHG}}$ | +2.797% | +1.712% | +0.803% | +6.467% | +7.925% | +5.535% | +3.521% | +3.116% | +1.935% |
| NGCF | 0.01115 | 0.04530 | 0.02805 | 0.00374 | 0.00958 | 0.00725 | 0.01570 | 0.04374 | 0.03168 |
| NGCF$_{\text{-UniHG}}$ | +3.139% | +4.327% | +4.528% | +13.636% | +12.735% | +11.310% | +4.713% | +8.802% | +6.787% |
| CSCF | 0.01540 | 0.06278 | 0.04142 | 0.00318 | 0.00596 | 0.00533 | 0.03992 | 0.11455 | 0.09264 |
| CSCF$_{\text{-UniHG}}$ | +14.675% | +17.776% | +27.764% | +6.918% | +31.543% | +9.005% | +0.601% | +1.422% | +1.759% |
| PSCF | 0.01536 | 0.06333 | 0.04247 | 0.00239 | 0.00475 | 0.00412 | 0.04119 | 0.11618 | 0.09437 |
| PSCF$_{\text{-UniHG}}$ | +15.169% | +17.116% | +21.427% | +7.531% | +5.684% | +7.282% | +0.048% | +3.408% | +3.157% |
| JSCF | 0.01899 | 0.07694 | 0.05477 | 0.00333 | 0.00661 | 0.00521 | 0.03770 | 0.11094 | 0.08975 |
| JSCF$_{\text{-UniHG}}$ | +1.421% | +1.546% | +2.921% | +19.365% | +29.728% | +25.432% | -1.405% | +0.252% | +1.225% |

Table 5: The average Recall and NDCG performance improvement rates after universal and domain-specific knowledge transfer. **Bold** represents better improvement.

| Methods | Amazon-book | | Yelp2018 | |
|---|---|---|---|---|
| | Recall@20 | NDCG@20 | Recall@20 | NDCG@20 |
| KGAT Wang et al. (2019a) | **+8.95%** | +10.05% | +7.18% | +5.54% |
| UniHG | +8.46% | **+11.48%** | **+17.52%** | **+11.71%** |

## 4.2 Recommendation Tasks for Knowledge Transfer

**Experiment Setting** To answer the Q4, we design a knowledge transfer scenario 'pre-training, retrieval of knowledge embeddings, self-supervised learning objectives' for UniHG and downstream tasks. Amazon-Book, Yelp2018, and Citeulike-a are used as downstream task datasets which inject universal knowledge. Specifically, we employ HGD to pre-train all node representations. Subsequently, multiple pre-trained knowledge embeddings will be retrieved for each item of the three downstream datasets. These embeddings will be considered as additional self-supervised signals in the recommendation tasks. For embedding retrieval, we collect raw text data of Amazon-Book, Yelp2018, and Citeulike-a. However, due to incomplete coverage of text data, performance might differ from other reported results. To ensure a fair comparison, we adopt the settings from He et al. (2020) for the Amazon Book and Yelp2018 datasets, and from Bogers & Van den Bosch (2008) for the CITEULIKE-a dataset. Please see Table 6 for the statistics of these datasets.

**Experiment Analysis** Table 4 and Table 5 show that: (i) The performance of all collaborative filtering methods improves with the introduction of universal knowledge. This indicates that universal semantic representations and out-domain interaction information between users and items can benefit downstream recommendation tasks. (ii) Compared to the domain-specific knowledge transfer approach employed by KGAT Wang et al. (2019a), UniHG achieves higher average recall enhancement and greater average NDCG improvement rates on both the Amazon-Book and Yelp datasets. This observation suggests that universal knowledge transfer may outperform domain-specific knowledge transfer due to the semantic consistency across domains. For instance, in a recommendation system, a new buyer without prior purchase history in a specific domain poses a challenge to domain-specific algorithms due to the 'cold start' problem Yin & Luo (2021). However, leveraging universal knowledge from additional prior information (e.g., user experience, profiles, or identity) may enhance preliminary recommendations. This implies that universal knowledge may provide more comprehensive information, which is often missing in domain-specific knowledge. We believe that to further tackle the task of cross-domain knowledge transfer, fostering the interaction between universal and domain-specific knowledge could be a promising research direction. This 'one for all' approach to universal knowledge transfer requires further exploration and remains the focus of our future work.

## 5 Conclusion

The absence of standardized evaluation protocols for **large-scale universal** heterogeneous graph analysis motivates our construction of UniHG – the first encyclopedic HG benchmark spanning multi-domain knowledge through a semantics-optimized automated pipeline. To address the representation learning challenges posed by UniHG's scale and attribute heterogeneity, we propose HGD, a framework featuring adaptive anisotropic feature propagation that dynamically adjusts propagation weights. Extensive benchmarks demonstrate the superiority: i) It achieves 24.1% accuracy gain on billion-edge subgraphs; ii) The learned representations show exceptional cross-domain adaptability, facilitating recommendation systems (11.4% NDCG gain) and link prediction (3.1% Acc. improvement) tasks through universal knowledge transfer.

## Acknowledgement

This work was supported by the National Natural Science Foundation of China (Grants No. 62476133) and the Fundamental Research Funds for the Central Universities (Grant No. 11300-312200502507).

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

# A   Notation

A graph can be defined as $\mathcal{G} = (\mathcal{V}, \mathcal{E})$, in which $\mathcal{V}$ and $\mathcal{E}$ represent the sets of nodes and edges, respectively. Each node $v_i \in \mathcal{V}$ and each edge $e_i \in \mathcal{E}$ is associated with a only one specific type. Formally, a heterogeneous graph (or heterogeneous information network) can be defined as follows:

$$\mathcal{G} = (\mathcal{V}, \mathcal{E}, \mathcal{A}, \mathcal{R}, \phi_v, \phi_e),$$

where:

- $\mathcal{V} = \{v_1, v_2, \ldots, v_N\}$ denotes the set of nodes, and $N$ is the total number of nodes.
- $\mathcal{E} = \{e_{ij}\}$ denotes the set of directed edges between nodes.
- $\mathcal{A} = \{a_1, a_2, \ldots, a_Q\}$ is the set of node classes, and $Q$ is the total number of node types.
- $\mathcal{R} = \{r_1, r_2, \ldots, r_M\}$ is the set of edge types, and $M$ is the total number of edge types.
- $\phi_v : \mathcal{V} \to \{a_1, a_2, \ldots, a_Q\}$ represents the node labeling function that maps entities to their corresponding classes.
- $\phi_e : \mathcal{E} \to \{r_1, r_2, \ldots, r_M\}$ represents the edge type function that maps edges to their corresponding types.

# B   Related Work

In this section, we review the previous works about HG datasets, then introduce works related to representative GNN methods.

## B.1   Heterogeneous Graph datasets

Various elevant works have been proposed Sharma et al. (2022); Chen et al. (2023); Ahrabian et al. (2023); Jang et al. (2022); Zhang et al. (2019) to construct domain-specific HG datasets, with the aim of facilitating research in the relevant fields. For instance, in the field of computer science, ACM Tang & Liu (2009) and DBLP Sun et al. (2011a) are utilized for classification tasks by leveraging computer-related literature information. IMDB Ma et al. (2011) is a movie dataset composed of information such as movies, actors, directors, and more. Its objective is to predict movie genre labels. LastFM Cantador et al. (2010) is a dataset sourced from the online music platform Last.fm, comprising users, artists, and artist tags, utilized for tasks such as link prediction. PubMed Sen et al. (2008) serves as a benchmark dataset for assessing heterogeneous graph embedding within biomedical literature, aiding researchers in gaining deeper insights into the heterogeneity of literature within the PubMed database. On the other hand, there are large-scale datasets such as Microsoft Academic Graph (MAG). The MAG is a heterogeneous graph containing records of scientific publications, consisting of billions of nodes and edges, making it at least an order of magnitude larger than the other CS (e.g., DBLP) and Med (e.g., Pubmed Kipf & Welling (2016)) academic datasets that are commonly used in existing heterogeneous GNN and heterogeneous graph mining studies. MAG provides a rich data resource for academic research and data-driven scientific analysis.

## B.2   Graph Neural Networks Methods

In recent years, HG methods are given more attention and can be divided into two categories. On one hand, meta-path-based methods have achieved success. Meng et al. Meng et al. (2015) pioneers the exploration of meta-paths to uncover meaningful patterns and dependencies in the graph structure. With the advent of GNNs, researchers integrate meta-path information into the learning process. Approaches such as MAGNN Fu et al. (2020) leverage GNN architectures to effectively propagate information along meta-paths, enabling the model to learn intricate patterns. On the other hand, aggregation-based methods Hu et al. (2020b); Veličković et al. (2017); Hong et al. (2020); Yang et al. (2021) for large-scale HGs employ sampling strategies similar to homogeneous graphs for aggregating each sampled subgraph. For instance, HGT Hu et al. (2020b) sets up multi-head transformers for modeling complex relationships between different node types on each relation. GAT Veličković et al. (2017) proposes to capture associations between different types through node-level attention mechanism. ieHGCN Yang et al. (2021) adopts the GCN framework and introduces inductive

neighbor sampling to enhance scalability on large-scale graphs. Meanwhile, HetSANN Hong et al. (2020) utilizes self-attention mechanisms to flexibly adjust the weights of node representations.

In contrast, our dataset is large-scale and generic-domain, and our method can adaptively mine multi-attribute association relationships without repeated sampling, achieving higher performance while reducing overhead.

# C  Theoretical Analysis

## C.1  Convergence Analysis of AFP

Consider the iterative propagation process:

$$\mathbf{C}^{k+1} = \text{Softmax}\left(\|\hat{\mathbf{A}} \times_2 \hat{\mathbf{C}}^k\|_2\right), \tag{7}$$

where the $\hat{\mathbf{A}} \in \mathbb{R}^{N \times N \times d}$ is the relation-aware adjacency tensor, the $\times_2$ denotes the tensor product, and the $\hat{\mathbf{C}}^k \in \mathbb{R}^{N \times N \times d}$ is the expanded feature tensor at step $k$.

Assume that the anisotropic feature propagation obeys the following theorem:

1. **Lipschitz Continuity**: The propagation operator $f(\hat{\mathbf{A}}, \mathbf{C}^k)$ is $L$-Lipschitz continuous ($L < 1$):
$$\|f(\hat{\mathbf{A}}, \mathbf{C}^k) - f(\hat{\mathbf{A}}, \mathbf{C}^{k-1})\|_F \leq L\|\mathbf{C}^k - \mathbf{C}^{k-1}\|_F, \quad \forall k \geq 1, \tag{8}$$

2. **Normalized Adjacency Tensor**: Each frontal slice $\hat{\mathbf{A}}_{:,:,i}$ satisfies:
$$\|\hat{\mathbf{A}}_{:,:,i}\|_2 \leq 1, \quad \forall i \in \{1, ..., d\}. \tag{9}$$

Under the above propagation process and assumptions, we can derive the convergence upper bound of AFP.

**Theorem 1 (Convergence Upper Bound)**  Under the above assumptions. the iterative sequence $\mathbf{C}^k$ converges exponentially to a unique fixed point $\mathbf{C}^*$ with:

$$\|\mathbf{C}^k - \mathbf{C}^*\|_F \leq \frac{L^k}{1-L}\|\mathbf{C}^0 - \mathbf{C}^1\|_F \tag{10}$$

**Proof**  For the relation-aware adjacency tensor $\hat{\mathbf{A}} \in \mathbb{R}^{N \times N \times d}$, each relation slice satisfies:

$$\|\hat{\mathbf{A}}_{:,:,m}\|_2 \leq \gamma < \frac{1}{\sqrt{2}}, \quad \forall m \in \{1, \dots, d\}, \tag{11}$$

where $\gamma$ is a predefined spectral radius upper bound. Using the submultiplicativity of Frobenius norm, we have:

$$\|\hat{\mathbf{A}} \times_2 \hat{\mathbf{C}}^k\|_F \leq \sum_{m=1}^d \|\hat{\mathbf{A}}_{:,:,m}\hat{\mathbf{C}}^k_{:,:,m}\|_F \leq \sum_{m=1}^d \|\hat{\mathbf{A}}_{:,:,m}\|_2\|\hat{\mathbf{C}}^k_{:,:,m}\|_F \leq \gamma\|\hat{\mathbf{C}}^k\|_F. \tag{12}$$

Then the linear propagation component $\mathbf{H}^{k+1} = \hat{\mathbf{A}} \times_2 \hat{\mathbf{C}}^k$ satisfies:

$$L_{\text{linear}} \triangleq \sup_{\hat{\mathbf{C}} \neq \hat{\mathbf{C}}'} \frac{\|\hat{\mathbf{A}} \times_2 (\hat{\mathbf{C}} - \hat{\mathbf{C}}')\|_F}{\|\hat{\mathbf{C}} - \hat{\mathbf{C}}'\|_F} \leq \gamma. \tag{13}$$

Based on the L2-Norm Non-expansiveness lemma, for any tensors $\mathbf{H}, \mathbf{H}' \in \mathbb{R}^{N \times N \times d}$, we have:

$$\|\|\mathbf{H}\|_2 - \|\mathbf{H}'\|_2\|_F \leq \|\mathbf{H} - \mathbf{H}'\|_F. \tag{14}$$

The composite nonlinear operation $\phi(\mathbf{H}) = \text{Softmax}(\|\mathbf{H}\|_2)$ satisfies:

$$L_{\text{nonlinear}} \triangleq \sup_{\mathbf{H} \neq \mathbf{H}'} \frac{\|\phi(\mathbf{H}) - \phi(\mathbf{H}')\|_F}{\|\mathbf{H} - \mathbf{H}'\|_F} \leq \sqrt{2}, \tag{15}$$

$$\|\phi(\mathbf{H}) - \phi(\mathbf{H'})\|_F \leq \sqrt{2}\|\|\mathbf{H}\|_2 - |\mathbf{H'}\|_2\|_F \leq \sqrt{2}\|\mathbf{H} - \mathbf{H'}\|_F. \tag{16}$$

Then the complete propagation operator $f(\hat{\mathbf{A}}, \mathbf{C}^k) = \phi(\hat{\mathbf{A}} \times_2 \hat{\mathbf{C}}^k)$ satisfies:

$$L_{\text{total}} \leq L_{\text{nonlinear}} \cdot L_{\text{linear}} \leq \sqrt{2}\gamma, \tag{17}$$

which constitutes a contraction mapping when $\gamma < \frac{1}{\sqrt{2}}$. Using the composition property of Lipschitz constants:

$$\|f(\mathbf{C}^k) - f(\mathbf{C'}^k)\|_F \leq L_{\text{nonlinear}}\|\hat{\mathbf{A}} \times_2 (\hat{\mathbf{C}}^k - \hat{\mathbf{C}}'^{\mathbf{k}})\|_F \leq L_{\text{nonlinear}} L_{\text{linear}}\|\hat{\mathbf{C}}^k - \hat{\mathbf{C}}'^{\mathbf{k}}\|_F \leq \sqrt{2}\gamma\|\mathbf{C}^k - \mathbf{C'}^{\mathbf{k}}\|_F \tag{18}$$

Thus for any $k \geq 1$, we have:

$$\|\mathbf{C}^{k+1} - \mathbf{C}^k\|_F = \|f(\hat{\mathbf{A}}, \mathbf{C}^k) - f(\hat{\mathbf{A}}, \mathbf{C}^{k-1})\|_F \leq L\|\mathbf{C}^k - \mathbf{C}^{k-1}\|_F, \tag{19}$$

where the $f(\cdot)$ forms a contraction mapping with factor $L < 1$. By Banach Fixed-Point theorem, there exists a unique $\mathbf{C}^*$ such that $\mathbf{C}^* = f(\hat{\mathbf{A}}, \mathbf{C}^*)$. Thus, for any $k \geq 0$, we have:

$$\|\mathbf{C}^k - \mathbf{C}^*\|_F \leq \sum_{i=k}^{\infty} \|\mathbf{C}^i - \mathbf{C}^{i+1}\|_F \leq \sum_{i=k}^{\infty} L^i \|\mathbf{C}^0 - \mathbf{C}^1\|_F = \frac{L^k}{1-L}\|\mathbf{C}^0 - \mathbf{C}^1\|_F. \quad \square \tag{20}$$

## C.2 Theoretical Guarantees for Heterogeneous Graphs

Considering the stability under edge perturbations, let $\Delta\mathbf{A}$ be a perturbation to the adjacency tensor. The perturbed propagation becomes:

$$\tilde{\mathbf{C}}^{k+1} = \text{Softmax}\left(\|(\hat{\mathbf{A}} + \Delta\mathbf{A}) \times_2 \hat{\mathbf{C}}^k\|_2\right). \tag{21}$$

**Theorem 2 (Topological Robustness)** For any perturbation $\Delta\mathbf{A}$ satisfying $\|\Delta\mathbf{A}\|_F \leq \epsilon$, the output perturbation $\Delta\mathbf{Z}_{\text{out}} = \|\mathbf{Z}_{\text{out}} - \tilde{\mathbf{Z}}_{\text{out}}\|_F$ is bounded by:

$$\|\Delta\mathbf{Z}_{\text{out}}\|_F \leq \epsilon \cdot \xi_{\text{Lip}} \sum_{k=0}^{K} \|\mathbf{\Theta}^k\|_2 + \mathcal{O}(\epsilon^2), \tag{22}$$

where $\xi_{\text{Lip}}$ is the Lipschitz constant of the encoder $\xi(\cdot)$.

**Proof** From the above assumption and Equ. 10, we have

$$\|\Delta\mathbf{C}^k\|_F = \|\tilde{\mathbf{C}}^k - \mathbf{C}^k\|_F \leq L^k\|\Delta\mathbf{A}\|_F. \tag{23}$$

The perturbed output of attention aggregation module can be calculated by:

$$\Delta\mathbf{Z}_{\text{out}} = \xi\left(\sum_{k=0}^{K} \mathbf{\Theta}^k(\mathbf{Z}^k + \Delta\mathbf{Z}^k)\right) - \xi\left(\sum_{k=0}^{K} \mathbf{\Theta}^k\mathbf{Z}^k\right) \leq \xi_{\text{Lip}} \sum_{k=0}^{K} \|\mathbf{\Theta}^k\|_2\|\Delta\mathbf{Z}^k\|_F \leq \xi_{\text{Lip}} \sum_{k=0}^{K} \|\mathbf{\Theta}^k\|_2 L^k\|\Delta\mathbf{A}\|_F. \tag{24}$$

Since $L < 1$, the geometric series converges is $\sum_{k=0}^{K} L^k \leq \frac{1}{1-L}$. Thus the first-order term scales linearly with $\|\Delta\mathbf{A}\|_F$.

## C.3 Complexity Analysis

For an undirected graph $\mathcal{G} = (\mathcal{V}, \mathcal{E})$, Kipf et al. Kipf & Welling (2016) demonstrated that in a Graph Convolutional Network (GCN), layer-wise message aggregation can be formulated as:

$$\mathbf{H}^{(l+1)} = \sigma\left(\tilde{\mathbf{D}}^{-\frac{1}{2}}\tilde{\mathbf{A}}\tilde{\mathbf{D}}^{-\frac{1}{2}}\mathbf{H}^{(l)}\mathbf{W}^{(l)}\right), \tag{25}$$

where $\tilde{\mathbf{A}} = \mathbf{A} + \mathbf{I}_N$, where $\mathbf{A}$ represents the adjacency matrix and $\mathbf{I}_N$ is the identity matrix. Thus, $\tilde{\mathbf{A}}$ is the adjacency matrix with added self-loops. The diagonal degree matrix $\tilde{\mathbf{D}}$ is defined as $\tilde{D}_{ii} = \sum_j \tilde{A}_{ij}$, where $\tilde{\mathbf{D}}$ represents the degree matrix of the vertices. $\mathbf{W}^{(l)}$ denotes the weight matrix

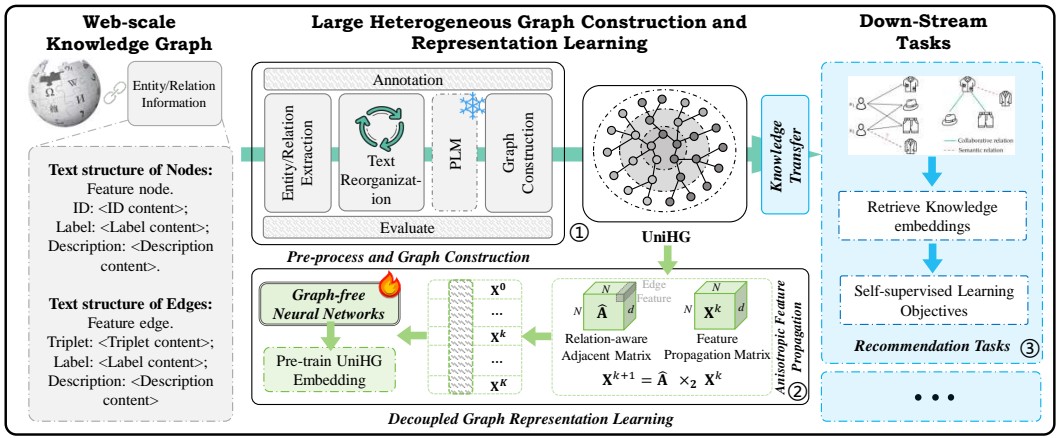

Figure 5: **Overall pipeline**. We pre-process structured knowledge from Wikidata and construct a large-scale heterogeneous graph through a series of operations (please see Figure 1 for detailed construction processes). Then, we propose a novel HGD framework to pre-train the UniHG node representations. Subsequently, we transfer the pre-trained knowledge embeddings from UniHG as universal knowledge to downstream recommendation system tasks.

of the $l$-th layer, $\sigma(\cdot)$ is the activation function, and $\mathbf{H}^{(l)} \in \mathbb{R}^{N \times d}$ represents the embedding matrix of the $l$-th layer. $N$ is the number of nodes, and $d$ is the hidden dimension. The time complexity of the convolution operation is $O(KN^2d)$, and the feature transformation has a complexity of $O(KNd^2)$, where $K$ is the number of convolutional layers. If $\mathcal{G}$ is a sparse graph, the convolution operation can be reduced to $O(K|E|)$ via sparse matrix multiplication, resulting in an overall time complexity of $O(K(|E| + Nd^2))$, where $|E|$ represents the number of edges. The $K$-layer hidden embeddings and adjacency matrix are stored on the GPU, and using sparse matrix multiplication can further reduce the computational cost on the GPU. The overall space complexity is $O(KNd + |E|)$. In our HGD framework, the feature propagation process can be expressed as:

$$\begin{cases} \mathbf{Z}^{k+1} = \zeta(Softmax(||\mathbf{H}^{k+1}||_2)), \\ \mathbf{Z}_{out} = \xi(\sum_{k=0}^{K}\mathbf{\Theta}^k\mathbf{Z}^k + \mathbf{XW}_r), \end{cases} \tag{26}$$

where $\xi$ denotes an encoder with a step connection matrix $\mathbf{W}_r$, $\mathbf{\Theta}^k$ represents the $k$-th diagonal attention matrix. The unfolding process involves replicating each row of matrix $\mathbf{C}$ and collecting them into multiple slices of a new three-dimensional tensor $\hat{\mathbf{C}}$. During the experiments, we adjusted the dimensions of $\mathbf{H}$, and then performed a row-wise softmax operation along the second dimension. The aforementioned formulation describes how matrix $\mathbf{H}$ is first normalized along the last dimension, followed by a dimension swap. The row-wise softmax is then applied to aggregate neighborhood information into the target node's features. This process is executed only once on the CPU, and the computational overhead of the fast tensor multiplication per iteration is negligible compare to full training. The resulting feature propagation matrix $\mathbf{Z}^k \in \mathbb{R}^{n \times d}$ undergoes multiple projection transformations on the GPU. In each layer of the HGD network, $\mathbf{Z}^k$ is only multiplied by a weight matrix $\mathbf{W}^{(l)} \in \mathbb{R}^{d \times d}$. Consequently, the time complexity and space complexity of HGD are $O(KNd^2)$ and $O(KNd)$, respectively.

In summary, using the HGD framework, the time complexity is reduced from $O(K|E| + KNd^2)$ to $O(KNd^2)$, and the space complexity is reduced from $O(|E| + KNd)$ to $O(KNd)$. This makes HGD particularly advantageous for handling large-scale graphs, as shown in Table 2 and Figure 7.

## D  More Experiments

### D.1  Baselines

We select 12 methods as baselines. Specifically, for the node classification task, on the one hand, we employ **GCN** Chen et al. (2018), **GAT** Veličković et al. (2017), **HGT** Hu et al. (2020b), and

Table 6: Statistics of three scales of the utilized graph datasets.

| Datasets | #Nodes | #Users | #Items | #Edges | #Classes | #Dimension | #Relations |
|---|---|---|---|---|---|---|---|
| UniHG-1M | 1,002,988 | - | - | 24,475,405 | 2,000 | 128 | 178 |
| UniHG-10M | 10,044,777 | - | - | 216,295,022 | 2,000 | 128 | 729 |
| UniHG-full | 77,312,474 | - | - | 564,425,621 | 2,000 | 128 | 2082 |
| Citeulike-a | - | 5,551 | 16,980 | 210,537 | - | 128 | 1 |
| Amazon-book | - | 52,643 | 65,865 | 2,090,149 | - | 128 | 39 |
| Yelp-2018 | - | 44,907 | 137,597 | 2,346,409 | - | 128 | 42 |

**MTMP** Pei et al. (2024) to aggregate neighbor information for each node on each sampled relation subgraph. On the other hand, we utilize **SGC** Wu et al. (2019), **SIGN** Frasca et al. (2020), and **GAMLP** Zhang et al. (2022c) to learn isotropic propagation features. Notably, the **HGT** is designed for large-scale HGs, while the others are not. To extend them to UniHG, we employ sampling and simplification techniques. Specifically, for **GCN** and **GAT**, we implement the HGSampling technique proposed by HGT Hu et al. (2020b). For **SGC**, **SIGN** and **GAMLP**, we apply the simplification designs proposed by OGB Hu et al. (2020a), treating UniHG as a homogeneous graph. For recommendation system tasks, we categorize popular model-based collaborative filtering methods into two types: graph-based methods **LightGCN** He et al. (2020) and **NGCF** Wang et al. (2019b), and sequence-based methods, **CSCF** Salton G (1983), **PSCF** Shardanand & Maes (1995), **JSCF** Jaccard (1901).

### D.2 Discussions of Meta-path-based Baselines

In this section, we discuss the challenges encountered when applying meta-path-based methods for representation learning on UniHG. First, meta-path-based approaches require the construction of sequences of node types connected by edge types, based on pre-defined recognition patterns. As illustrated in Figure 6, if the recognition pattern is set as (Red $\rightarrow$ Blue $\rightarrow$ Red $\rightarrow$ Blue), for a single-relation graph (left), a path $L$ of length 4 starting from node $n_1$ can generate three meta-paths. However, for a multi-relational graph (right), the same path $L$ can generate 57 meta-paths. As the number of relation types increases, the number of meta-paths grows ex-

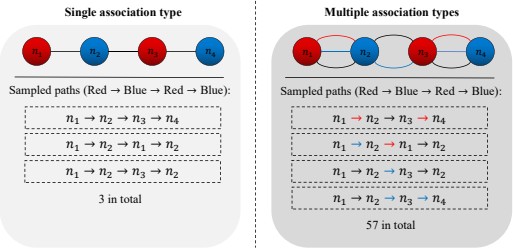

Figure 6: Examples of meta-path construction. Single relation type graph situation (left) and multiple relation type graph situation (right).

ponentially, which poses significant challenges for meta-path-based methods when applied to UniHG, especially in scenarios where relation types cannot be ignored. We take **SeHGNN** Yang et al. (2023) as an example to illustrate the encountered out-of-memory (OOM) issue. SeHGNN has demonstrated excellent performance on the OGBN-MAG dataset. However, due to the complexity of UniHG (77.31 million nodes, 564 million edges, and 2,082 relation types), applying SeHGNN to UniHG incurs substantial time and space costs, far exceeding those required for OGBN-MAG (1.9 million nodes, 21 million edges, and 7 relation types). SeHGNN propagates node features along meta-paths to compute node feature matrices for each meta-path view. The space complexity of this process is $O(RNd)$, where $R$ is the number of meta-path types. In MAG, $R = 7$ (considering only second-order meta-paths), while in UniHG, $R = 2,082$ (similarly considering second-order meta-paths). $N$ is the number of nodes, which is 77,312,474 for UniHG, and $d = 256$ is the node feature dimension, consistent with MAG. Under this setup, the memory required to compute the heterogeneous meta-path features of UniHG is approximately $2082 \times 77312474 \times 256 \times 2B \approx 74.95$ TB (assuming float16). Such a high space overhead is impractical. In contrast, the proposed HGD incurs approximately 270 GB of memory when training on UniHG, which is over 284 times smaller than SeHGNN (and in practice, the difference is even larger, as we only compared the feature matrix size for SeHGNN). Furthermore, we evaluate the representation learning capability of the meta-path-based method ieHGCN Yang et al. (2021) on the UniHG-1M dataset, ignoring the edge-type information. As shown in Table 7, after the same number of iterations, ieHGCN achieved an accuracy of 40.47%, a

recall of 41.88%, and an F1-score of 37.18. These results, along with the discussions, demonstrate that meta-path-based methods are not well-suited for large-scale heterogeneous graphs like UniHG. Additionally, we experiment with methods not designed for large-scale heterogeneous graphs, such

Table 7: Results compared to ieHGCN on UniHG-1M. The best performances are highlighted in **bold**.

| Methods | Accuracy | Recall | F1 Score |
|---|---|---|---|
| ieHGCN Yang et al. (2021) | 40.47 | 41.88 | 37.18 |
| **HGD (ours)** | **75.41** | **75.95** | **82.64** |

Table 8: Hyperparameters for main experiments.

| Methods | num_hops | layers | n_layers_1 | num_heads | hidden_channels | lr | epochs | $\alpha$ | seed |
|---|---|---|---|---|---|---|---|---|---|
| GCN | - | 3 | - | 8 | 256 | 0.01 | 300 | - | 2025 |
| GAT | - | 3 | - | 8 | 256 | 0.01 | 300 | - | 2025 |
| HGT | - | 3 | - | 8 | 256 | 0.01 | 300 | - | 2025 |
| SGC | - | 3 | - | - | 256 | 0.01 | 300 | - | 2025 |
| SIGN | 4 | 3 | - | - | 256 | 0.01 | 300 | - | 2025 |
| GAMLP | 5 | 3 | 1 | 1 | 256 | 0.01 | 300 | 0.5 | 2025 |
| HGD | 5 | 3 | - | 1 | 256 | 0.01 | 300 | - | 2025 |

Table 9: Hyperparameters for recommendation tasks.

| Methods | regs | embed_size | layers | lr | batch_size | epochs | node_drop | mess_drop | recons | decay | recdim | topks | seed |
|---|---|---|---|---|---|---|---|---|---|---|---|---|---|
| LightGCN | - | - | 3 | 1e-3 | - | - | - | - | - | 1e-4 | 256 | 20 | 2025 |
| NGCF | 1e-5 | 64 | 3 | 5e-4 | 1024 | 220 | 0.1 | 0.1 | True | - | - | - | 2025 |

as **HINormer** Mao et al. (2023). Detailed observations are as follows: HINormer is not designed for large-scale heterogeneous graphs, making its application in such contexts challenging. HINormer utilizes breadth-first search (BFS) to generate node sequences as input to the Transformer model. In large-scale graphs, high average degrees result in nodes being predominantly located within the 1-hop neighborhood of the target node. This leads to over-smoothing of node features after message passing. When training a Transformer with overly long sequences, attempts to alleviate over-smoothing by increasing the length of random walk sequences introduce significant time and space overheads, along with convergence difficulties.

### D.3    Semi-supervised Experiments

Considering that semi-supervised learning may be more practical for real-world applications, we further evaluate HGD and other graph decoupling baseline methods on UniHG-1M under a semi-supervised setup (by removing 30% of the labels compared to the fully-supervised training set). From the results in the table 10, it can be observed that the proposed HGD method consistently achieves performance improvements, even with reduced label availability.

### D.4    Link Prediction Experiments under the Domain Transfer Setting

To rigorously validate the cross-domain generalization capacity of our proposed methodology, we conducted systematic benchmarking across heterogeneous graph domains following the experimental protocols established in References Chamberlain et al. (2022); Zhang & Chen (2018). Our evaluation encompasses five representative datasets spanning distinct network typologies: citation networks (Cora, Citeseer, PubMed), collaborative networks (ogbl-collab), and biomedical interaction graphs (ogbl-ddi). As detailed in Table 11, the implemented UniHG framework demonstrates consistent performance enhancements through domain-adaptive transfer learning.

Table 10: Results of Semi-supervised comparison experiments on UniHG-1M. The best performances are highlighted in **bold**.

| Methods | Accuracy | Recall | F1 Score |
|---|---|---|---|
| SGC Wu et al. (2019) | 43.39 | 42.61 | 57.12 |
| SIGN Frasca et al. (2020) | 52.56 | 50.12 | 65.71 |
| GAMLP Zhang et al. (2022b) | 33.37 | 31.73 | 46.37 |
| **HGD (ours)** | **70.16** | **72.53** | **77.01** |

Table 11: Results of link prediction experiments under the domain transfer setting.

| Method | CORA | Citeseer | Pubmed | ogbl-collab | ogbl-ddi |
|---|---|---|---|---|---|
| BUDDY Chamberlain et al. (2022) | 84.50 | 89.52 | 69.56 | 68.49 | 79.62 |
| BUDDY$_{-UniHG}$ | **85.39** | **91.02** | **72.65** | **68.97** | **79.97** |
| ELPH Chamberlain et al. (2022) | 86.28 | 88.80 | 73.56 | OOM | 31.64 |
| ELPH$_{-UniHG}$ | **86.38** | **89.39** | 73.42 | OOM | **31.99** |
| SEALGIN Zhang & Chen (2018) | 72.26 | 75.33 | 64.02 | – | – |
| SEALGIN$_{-UniHG}$ | **72.56** | **76.19** | **65.25** | – | – |
| SEALGCN Zhang & Chen (2018) | 72.05 | 74.56 | 65.83 | – | – |
| SEALGCN$_{-UniHG}$ | **72.75** | **76.19** | **66.91** | – | – |
| SEALSAGE Zhang & Chen (2018) | 67.42 | 76.14 | 60.39 | – | – |
| SEALSAGE$_{-UniHG}$ | **70.09** | **76.87** | **61.03** | – | – |

## D.5 Runtime Experiments

To quantitatively benchmark the computational efficiency of our proposed Heterogeneous Graph Diffusion (HGD) framework, we theoretical compare the complexity, and conduct rigorous runtime analyses under standardized multi-label node classification settings.

**Complexity Comparison** For large-scale graphs, the key contribution of feature propagation is to reduce complexity. Our AFP module further proposes an anisotropic design for heterogeneous data to handle large-scale heterogeneous graphs with complex attributes containing more than 2000 edge types. Table 12 shows the comparison of time and space complexity and heterogeneity processing.

Table 12: Computational complexity comparison of heterogeneous graph processing paradigms (N: nodes, E: edges, d: feature dimension, K: layers, R: relation types)

| Method | Time Complexity | Space Complexity | Heterogeneity Handling |
|---|---|---|---|
| Global Convolution | $O(K(RE+RNd))$ | $O(E+KNd)$ | Meta-path dependent |
| Vanilla Propagation | $O(KNd)$ | $O(KNd)$ | Isotropic |
| **AFP** (Ours) | $O(KNd)$ | $O(KNd)$ | **Relation-aware anisotropic** |

**Runtime Analyses** As depicted in Figure 7 and Table 13, our evaluation encompasses three scaled UniHG datasets ($10^6$–$10^9$ nodes). Key findings reveal fundamental limitations in existing paradigms: i) HGT exhibits prohibitive quadratic time complexity $O(R^2)$ with respect to relation types (R=2082 in UniHG-Full), requiring 108.58±8.71 second per epoch for million-node graphs (mean ± std. dev., 5 runs). ii) HGD demonstrates sublinear scaling $O(R)$ via our diffusion operator and achieves 2.8–22.1× acceleration over HGT. The empirical evidence establishes HGD as a effective solution for large-scale heterogeneous graph processing, achieving an balance between computational tractability (38.78s/epoch for $10^6$ nodes) and model fidelity (96.09 F1 score).

## D.6 Dynamic Graph Experiments

To rigorously evaluate the dynamic adaptation capabilities of our **HGD** framework, we conducted temporal graph benchmarking following the standardized evaluation protocol of Temporal Graph Benchmark (TGB) Huang et al. (2023). As shown in Table 14, **HGD** achieves statistically significant

Table 13: Runtime comparison four baseline methods on three datasets.

| Method | UniHG-1M | UniHG-10M | UniHG-Full |
|---|---|---|---|
| HGT | 108.58±8.71s | 1,074.33±82.15s | 4,032.58±318.72min |
| SIGN | 38.54±10.61s | 206.98±28.62s | 161.32±14.91min |
| GAMLP | 39.49±7.91s | 227.98±44.25s | 177.76±16.40min |
| **HGD** (Ours) | 38.78±8.98s | 233.55±46.55s | 182.47±21.99min |

improvements with 2.4% and 1.8% relative gains in test/validation NDCG over two competitive baselines, demonstrating superior temporal pattern capture.

Table 14: Results of temporal node classification experiments.

| Method | Test NDCG | Val. NDCG |
|---|---|---|
| DyRep Trivedi et al. (2019) | 0.374 | 0.394 |
| TGN Rossi et al. (2020) | 0.374 | 0.395 |
| **HGD** (Ours) | **0.383** | **0.401** |

## D.7 Robustness Analysis of HGD

To systematically evaluate the robustness of our **Heterogeneous Graph Diffusion (HGD)** framework under structural perturbations, we conducted a controlled node masking study on the **UniHG-1M** dataset (1M nodes, 178 edge types).

Table 15: Node classification performance under progressive structural perturbations (mean ± std. dev. over 5 runs)

| Masking Ratio | 10% | 20% | 30% | 40% | 50% | 60% | 70% |
|---|---|---|---|---|---|---|---|
| Accuracy (%) | 75.26±0.32 | 75.02±0.28 | 75.80±0.41 | 75.72±0.37 | 76.04±0.35 | 75.74±0.39 | 75.42±0.43 |
| Recall (%) | 75.92±0.29 | 76.59±0.31 | 76.72±0.33 | 76.43±0.35 | 76.76±0.38 | 76.27±0.40 | 76.01±0.42 |
| F1-score (%) | 82.65±0.25 | 82.14±0.27 | 82.85±0.30 | 83.06±0.32 | 83.31±0.29 | 83.03±0.34 | 82.94±0.37 |

The experimental results are shown in Table 15, which shows that HGD can maintain a stable accuracy under severe perturbations, proving the effectiveness and robustness of the AFP module. We found that shielding some nodes can improve generalization ability.

## D.8 Statistical Significance Analysis

We conducted rigorous empirical validation through 10 independent trials with controlled random seeds (Fig. 17). HGD demonstrates superior stability with minimal performance variance ($\sigma$=0.3% vs. baseline $\sigma$=0.7-1.2%), quantified by the relative standard deviation metric:

$$RSD = \frac{\sigma}{\mu} \times 100 \qquad (27)$$

where HGD achieves RSD=0.4% on UniHG-1M, outperforming baselines by 3.7-4.9×. **Two-tailed t-tests** with Bonferroni correction confirm statistical significance ($p<0.001$) across all comparisons (Table 16). The large effect sizes (**Cohen's d**>1.89) substantiate the practical significance of improvements.

## D.9 Hyperparameters and Environments

To maintain fairness, the hyperparameter settings are kept consistent for all methods on each dataset. For UniHG, we use 5-hops anisotropic propagation to generate aggregated features as the input for model training. Then, we employ the 18 layers hidden network as the graph encoder, which transforms the feature dimension of the input graph from 128 to 256. Besides, the learning rate is 0.01 and the batch size is 300,000 for UniHG-full, 100,000 for UniHG-10M, 20,000 for UniHG-1M. On the other hand, for the Amazon-Book, Yelp2018, and Citeulike-a datasets, we use three-layer

Table 16: Statistical significance analysis on UniHG datasets

| Comparison | Dataset | Cohen's d | p-value |
|---|---|---|---|
| SGC vs. HGD | UniHG-1M | 2.74 | <0.0001 |
| GAMLP vs. HGD | UniHG-1M | 3.12 | <0.0001 |
| SIGN vs. HGD | UniHG-1M | 1.89 | 0.0013 |

Table 17: Performance stability across random seeds

| Method | Dataset | Mean Acc (%) | Std Dev |
|---|---|---|---|
| SGC | UniHG-1M | 42.5 | 0.7 |
| GAMLP | UniHG-1M | 44.5 | 0.8 |
| SIGN | UniHG-1M | 56.7 | 1.2 |
| HGD | UniHG-1M | 75.4 | 0.3 |

LightGCN and NGCF as the feature encoder to learn the representation of users and items. The dimension of the hidden feature is 256, the learning rate is 0.001, and the batch size is 204. All experiments were conducted using a single 24GB GeForce RTX 4090 GPU. The hyperparameters of our main experiments and recommendation tasks for konwledge transfer can be found in Table 8 and Table 9.

# E  Advantages over Knowledge Graph Datasets

Table 18: Comparative analysis between conventional knowledge graphs and UniHG.

| Aspect | Knowledge Graphs | UniHG | Key Advantages |
|---|---|---|---|
| Feature Representation | Discrete triples | Dense embeddings | Compact feature encoding |
| Storage Efficiency | 1.7 TB | 37.5 GB | 45:1 compression ratio |
| Dynamic Scalability | Weekly/Monthly updates | Incremental updates (<2 min) | Faster update |
| Construction Cost | Triple (human annotation) | Triple (automated pipeline) | Cost reduction |
| Cross-domain Transfer | Domain-specific architectures | Unified embedding space | 11.48% higher NDCG@20 |

**Comparative Analysis.** We systematically contrast our UniHG framework with conventional knowledge graphs through five critical dimensions (Table 18). First, the transition from symbolic triples to low-dimension embeddings enables *lossless compression* of semantic features, achieving parameter reduction while preserving relational semantics. Second, the compressed embedding space demonstrates remarkable storage efficiency, condensing typical 1.7 TB knowledge graphs into 37.5 GB representations (45:1 compression ratio) through tensor factorization techniques. The dynamic updating mechanism exhibits orders-of-magnitude improvement, reducing update latency from weekly-level to sub-2-minute incremental updates. Most notably, the unified embedding space enables cross-domain knowledge transfer.

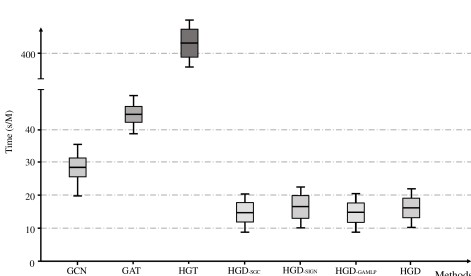

Figure 7: Training time statistics of baselines for multi-label node classification. We report the time for training each 1M nodes. (Low is faster)

Benchmarking on the Amazon-book dataset shows 11.48% relative NDCG@20 improvement compared to domain-specific baselines, validating our design's compatibility with pretrain-finetune paradigms.

**Dataset Scalability.** Currently, UniHG only leverages knowledge from Wikidata. But the proposed overall pipeline can be seamlessly extended to other knowledge graphs (such as Freebase, OpenCyc). We also provide open source code to promote related research. In future work, we will try to expand to more knowledge sources.

# F  Limitation

Currently, although the proposed overall pipeline can be seamlessly extended to other knowledge graphs, UniHG only leverages knowledge from Wikidata. Thus supplementary other knowledge bases as reference or new content may improve the quality of the proposed dataset.

