# OpenReview forum: "UniHG: A Large-scale Universal Heterogeneous Graph Dataset and Benchmark for Representation Learning and Cross-Domain Transferring"
_NeurIPS.cc/2025/Datasets_and_Benchmarks_Track — NeurIPS 2025 Datasets and Benchmarks Track poster_

### Official Review · Reviewer_24pH · 2025-06-30

**Rating:** 4
**Confidence:** 4

**Summary:**

This paper presents UniHG, a large-scale and universal heterogeneous graph (HG) dataset derived from Wikidata, comprising over 77.31 million nodes and 564 million edges across 2,082 relation types. Alongside the dataset, the authors propose HGD (Heterogeneous Graph Decoupling), a new representation learning framework designed to enable scalable and effective learning on multi-attribute, multi-hop HGs. Extensive experiments on node classification and recommendation tasks are conducted to demonstrate the scalability and generalization of the proposed methods.

This paper addresses an important need in the HG research community: the lack of large-scale, cross-domain benchmarks. However, the evaluation is somewhat narrow in scope, and the generalizable conclusions and some methodological choices require deeper empirical and theoretical validation.

**Additional Feedback:**

None

**Dataset Code Accessibility:**

Partly

**Dataset Code Comments:**

Lack of complete documentation.

**Ethical Considerations:**

No, there are no or only very minor ethics concerns

**Final Justification:**

All the questions are clear addressed.

**Limitations Weaknesses:**

1. The scope of downstream task verification is limited, and the conclusion of generality is still weak.
Although the author claims that UniHG has "general knowledge transfer capabilities", the experiment only covers 3 recommendation system datasets (Amazon, Yelp, Citeulike), and does not show the applicability of broader tasks such as graph classification, relationship prediction, and graph generation. In addition, the failure to cover realistic and complex tasks (such as financial fraud detection, biomedical relationship mining, social graph recommendation, etc.) weakens the persuasiveness of the "cross-domain general graph benchmark".

2. The label aggregation strategies have not been sufficiently evaluated.
The original labels were reduced from 74,666 to 2,000, which is a very aggressive scale compression, but the article lacks quantitative analysis of semantic consistency and information loss and discussion of label merging errors or redundancy

Although the author used GPT-4 and manual inspection ($1 per mapping), quantitative indicators such as consistency ratio, conflict rate, and final retention ratio were not reported.

3. An in-depth comparison with existing SOTA methods has not been provided, such as:
(1) Zhao Z, Liu Z, Wang Y, et al. RA-HGNN: Attribute completion of heterogeneous graph neural networks based on residual attention mechanism[J]. Expert Systems with Applications, 2024, 243: 122945.
(2) Liu Z, Xie M, Song Y, et al. BAB-GSL: Using Bayesian influence with attention mechanism to optimize graph structure in basic views[J]. Neural Networks, 2025, 181: 106785.
(3) Zou Y, Fang Z, Wu Z, et al. Revisiting multi-view learning: A perspective of implicitly heterogeneous Graph Convolutional Network[J]. Neural Networks, 2024, 169: 496-505.

4. The theoretical support and interpretability of the AFP module are still weak.
Although anisotropic feature propagation mechanism was proposed, there is a lack of explanatory analysis of propagation mode and information flow modeling capabilities.

**Strengths Contributions:**

1. A HG Dataset and Scalable Construction Pipeline
The largest known heterogeneous graph dataset (UniHG) has been constructed, containing over 77 million nodes, 564 million edges, and more than 2,000 relation types. A gap in existing HG benchmarks—often limited to small-scale or domain-specific settings—has been addressed. The dataset has been built from Wikidata through a carefully designed, automated, and semantically aligned pipeline, in which novel strategies for entity/relation filtering, semantic alignment, and label consolidation (reducing 74,000+ categories to 2,000) have been applied.
2. A Representation Learning Framework (HGD)
A new graph representation learning framework (HGD) has been proposed, in which Anisotropic Feature Propagation (AFP) and graph-free mapping strategies are incorporated to enable efficient and scalable learning on large heterogeneous graphs.
3. The Potential of Cross-Domain Transfer
The potential for cross-domain knowledge transfer has been demonstrated through recommendation tasks, where pre-trained embeddings from UniHG have been utilized to improve collaborative filtering models, leading to measurable gains in NDCG and Recall across multiple datasets.
4. Open Benchmark & Reproducibility
Both the dataset and source code have been released, offering significant value to the research community and supporting standardization and reproducibility in heterogeneous graph learning.

---

> ### Author Rebuttal · Authors · 2025-07-31
>
> Thank you for your positive feedback on our largest dataset, new graph representation learning framework, improved performance, comprehensive analyses, significant value to the research community. Regarding your concerns, we provide the following responses below.
>
> ### **Q1**: The scope of downstream task verification is limited, and the conclusion of generality is still weak. Although the author claims that UniHG has "general knowledge transfer capabilities", the experiment only covers 3 recommendation system datasets (Amazon, Yelp, Citeulike), and does not show the applicability of broader tasks such as graph classification, relationship prediction, and graph generation. In addition, the failure to cover realistic and complex tasks (such as financial fraud detection, biomedical relationship mining, social graph recommendation, etc.) weakens the persuasiveness of the "cross-domain general graph benchmark".
>
>
> **A1**: Thanks for your comment. For the verification on downstream tasks, we conduct the experiments including: node classification, recommendation system task, link prediction, and dynamic graph classification. The first two tasks are presented in the main body of the paper, while the latter two are given in the Appendix. These experiments basically address the cases you mentioned, for examples: i) biomedical relationship mining on ogbl-ddi in Table 11 of Appendix, ii) social graph relationship mining on ogbl-collab in Table 11 of Appendix, iii) node classification in Table 10 of Appendix, iv) node classification on dynamic graph in Table 14 of Appendix, v) link prediction on five datasets in Table 11 of Appendix. Thus, we believe these experiments are enough to demonstrate the effectiveness of our benchmark.
>
>
> ### **Q2**: The label aggregation strategies have not been sufficiently evaluated. The original labels were reduced from 74,666 to 2,000, which is a very aggressive scale compression, but the article lacks quantitative analysis of semantic consistency and information loss and discussion of label merging errors or redundancy. Although the author used GPT-4 and manual inspection ($1 per mapping), quantitative indicators such as consistency ratio, conflict rate, and final retention ratio were not reported.
>
>
> **A2**: Thanks for your comment. Regarding your concerns about label generation, first, we clarify our processing strategies below.
>
> i) **Manual Checking**: The cluster descriptions generated by pretrained models underwent rigorous manual review, for example, per mapping (from sample to cluster) was checked by at least 3 annotators (please see Section 2.4 in the manuscript). The human verification can largely reduce potential noise/bias.
>
> ii) **Prompt Engineering**: Prompt templates could enforce structured outputs (please refer to the text reorganization module in Figure 1) , enhancing noise resilience.
>
> According to your suggestion, next, we provide quantitative metrics for this process:
>
> i) **Consistency Ratio**: The consistency score exceeds 85%, following the rule that all annotators consistently agreed on each cluster-label mapping.
>
> ii) **Conflict Rate**: The conflict rate is below 3%, determined by the rule that at least two annotators must disagree on a cluster-label mapping.
>
> iii) **Retention Ratio**: The final retention ratio exceeds 98.4% (73,529/74,666) on original raw labels after removing controversial labels through manual checking.
>
>
> ### **Q3**: An in-depth comparison with existing SOTA methods has not been provided, such as: (1) Zhao Z, Liu Z, Wang Y, et al. RA-HGNN: Attribute completion of heterogeneous graph neural networks based on residual attention mechanism[J]. Expert Systems with Applications, 2024, 243: 122945. (2) Liu Z, Xie M, Song Y, et al. BAB-GSL: Using Bayesian influence with attention mechanism to optimize graph structure in basic views[J]. Neural Networks, 2025, 181: 106785. (3) Zou Y, Fang Z, Wu Z, et al. Revisiting multi-view learning: A perspective of implicitly heterogeneous Graph Convolutional Network[J]. Neural Networks, 2024, 169: 496-505.
>
>
> **A3**:  Thanks for your comment. Regarding the three baselines you mentioned, we add a comparison with RA-HGNN [Zhao et al., 2024] (see results below), while other two methods were excluded due to their prohibitive computational demands (see the following explanation).
>
> - Results of RA-HGNN [Zhao et al., 2024] on UniHG:
> Acc (1M) 55.60, Recall (1M) 55.35, F1 Score (1M) 58.17 Acc (10M) 56.60, Recall (10M) 58.99, F1 Score (10M) 51.75, Acc (Full) 54.87, Recall (Full) 56.64, F1 Score (Full) 58.80. The RA-HGNN performance is largely lower than our method.
>
> - Prohibitive computation cost for BAB-GSL [Liu et al., 2025] and Revisiting Multi-view Learning [Zou et al., 2024]. BAB-GSL is computationally infeasible for UniHG containing over 77.3M nodes, due to the use and calculation on dense adjacency-matrix views. The latter method (Revisiting Multi-view Learning) is a meta-path method, which is not scalable to UniHG due to the explosion of meta-path numbers (requiring about 74.95TB memory, please see the discussion in the appendix §D.2).
>
>
> ### **Q4**: The theoretical support and interpretability of the AFP module are still weak. Although anisotropic feature propagation mechanism was proposed, there is a lack of explanatory analysis of propagation mode and information flow modeling capabilities.
> ﻿
> **A4**: Thanks for your suggestion. We appreciate your feedback and would like to clarify that the theoretical supports are provided both in the mainbody part and in the Appendix C.
>
> - **Convergence Upper Bound of AFP**:
>
> Under the assumptions of Lipschitz continuity and a normalized adjacency tensor, the iterative sequence $\mathbf{C}^k$ converges exponentially to a unique fixed point $\mathbf{C}^*$ with the bound:
>
> $\left|\left|\mathbf{C}^k-\mathbf{C}^*\right|\right\|_F \leq \frac{L^k}{1-L}\left|\left|\mathbf{C}^0-\mathbf{C}^1\right|\right|_F$, (Eq. 10)
>
> where $L < 1$ is the Lipschitz constant of the propagation operator $f(\hat{\mathbf{A}}, \mathbf{C}^k)$. Please refer to Appendix C.1 for the concrete derivation and analysis.
>
> - **Topological Robustness**:
>
>
> The perturbation in the final output representation $\Delta\mathbf{Z}\_{\text{out}} = \|\|\mathbf{Z}\_{\text{out}} - \tilde{\mathbf{Z}}\_{\text{out}}\|\|\_{F}$ is bounded by:
>
> $\|\|\Delta\mathbf{Z}\_{\text{out}}\|\|\_{F} \leq \epsilon \cdot \xi\_{\text{Lip}} \sum\_{k=0}^{K} \|\|\boldsymbol{\Theta}^{k}\|\|\_{2} + \mathcal{O}(\epsilon^{2})$, (Eq. 22)
>
> where $\xi\_{\text{Lip}}$ is the Lipschitz constant of the feature mapping encoder $\xi(\cdot)$, and $\boldsymbol{\Theta}^{k}$ denotes the $k$-th diagonal attention matrix in the hop-wise aggregation. The concrete derivation and analysis was given in Appendix C.2, Pages 15.
>
>
> We appreciate your feedback and ensure further clarifications in the next revision.

---

> > ### Comment · Reviewer_24pH · 2025-08-06
> > **Thank you for your response.**
> >
> > Thank you for the caerfully responses. It's much clearer now. I would like to raise my score to borderline accept.

---

> > > ### Author Response · Authors · 2025-08-06
> > >
> > > Thanks for your positive feedback and for confirming the clearer explanation regarding the experimental scope and algorithm theory. We appreciate your time and valuable comments once again.

---

### Official Review · Reviewer_F2Gq · 2025-07-01

**Rating:** 5
**Confidence:** 3

**Summary:**

This paper constructs a large-scale universal heterogeneous graph dataset called UniHG, which could facilitate heterogeneous graph representation learning and cross-domain knowledge mining. The dataset is constructed from Wikidata and follows the pipeline of ‘Entity/Relation Extraction and Filtering - Semantic Alignment - Node Annotation’. In addition to the dataset, the paper also presents a heterogeneous graph decoupling framework for representation learning and achieves improved performance over baselines.

**Dataset Code Accessibility:**

Yes

**Dataset Code Comments:**

The dataset and code are easily accessible.

**Ethical Considerations:**

No, there are no or only very minor ethics concerns

**Final Justification:**

The authors have addressed my concerns, and I keep my positive score.

**Limitations Weaknesses:**

1. The figures are not properly ordered. For example, Figure 5 should be placed in the introduction section, and Figure 2 should be replaced by Figure 3 so that the figure can be cited in Sec. 2.5.
2. The constructed dataset only has one node type. Will this limit the application of the dataset, as other datasets have more types?
3. Why use CLIP as the feature projector? The authors are encouraged to discuss other alternative LLMs.
4. When using cross-entropy as the training objective, is the ground-truth label the original label or the cluster label?

**Strengths Contributions:**

1. The constructed dataset is large-scale and would facilitate the research of heterogeneous graph learning.
2. The code and dataset are easily accessible.
3. The proposed method for heterogeneous graph learning achieves improved performance over the baselines.
4. Comprehensive theoretical analyses are provided.

---

> ### Author Rebuttal · Authors · 2025-07-30
>
> Thanks for your positive comments on our large-scale datasets, easily accessible code and dataset, improved performance, and comprehensive analyses. Regarding your concerns, we make the responses in detail below.
>
> ### **Q1**: The figures are not properly ordered. For example, Figure 5 should be placed in the introduction section, and Figure 2 should be replaced by Figure 3 so that the figure can be cited in Sec. 2.5.
>
> **A1**: Thanks for your comment. We make sure to revise them in the version, including moving Figure 5 to the corresponding position of Section 1, adjusting the placement of Figures 2 and 3 for better readability.
>
>
> ### **Q2**: The constructed dataset only has one node type. Will this limit the application of the dataset, as other datasets have more types?
>
> **A2**: Thanks for your comment. The dataset indeed contains different-attribution nodes with different text descriptions. Node features are initially generated from text that include attribution description, and further optimized through graph learning. As a result, the node features inherently capture implicit entity type information (see Section 2.2), rather than relying on explicit node type distinctions. Adopting a universal node representation enhances the dataset’s applicability by reducing semantic gaps across different graph datasets. The improvement (see Table 4) on the downstream tasks also verify this point.
>
>
> ### **Q3**: Why use CLIP as the feature projector? The authors are encouraged to discuss other alternative LLMs.
>
> **A3**: Thanks for your suggestion. CLIP is indeed a widely used models for text feature extraction especially for label information. And, CLIP is lightweight and easy to use. Of course, others LLMs such as GPT could be used. But evaluating different LLMs for feature extraction is beyond the scope of this work. And due to time limitation during the rebuttal phase, we defer the concrete experimental discussion to the next revision.
>
>
> ### **Q4**: When using cross-entropy as the training objective, is the ground-truth label the original label or the cluster label?
>
> **A4**: The ground-truth label is cluster label checked manually, not the original label.

---

> > ### Comment · Reviewer_F2Gq · 2025-08-03
> >
> > Thanks for the clarification, and I keep my positive score.

---

> > > ### Author Response · Authors · 2025-08-03
> > >
> > > Thanks for your positive feedback on our clarification. We appreciate your time and valuable comments again.

---

### Official Review · Reviewer_faTf · 2025-07-01

**Rating:** 4
**Confidence:** 4

**Summary:**

The manuscript constructs a large-scale heterogeneous graph dataset extracted from Wikidata. The dataset would advance the research on the heterogeneous graph. The manuscript also proposed a novel representation learning framework, Heterogeneous Graph Decoupling (HGD), which shows strong performance on the proposed benchmark.

**Dataset Code Accessibility:**

Yes

**Dataset Code Comments:**

Code and dataset are available.

**Ethical Comments:**

N.A.

**Ethical Considerations:**

No, there are no or only very minor ethics concerns

**Final Justification:**

The authors' response and additional have well-solved my concerns. It indicates that using graph to model the data is meaningful. Based on this, I would like to support the acceptance of the paper.

**Limitations Weaknesses:**

My main concerns are the motivation behind this paper—why do we need to construct a heterogeneous graph? Is using a graph necessary for the same task? What are the advantages of structuring the data as a graph compared to the original text data? Additionally, the experimental results of this dataset on LLMs with text data should also be reported.

**Strengths Contributions:**

1. The manuscript constructs a very large-scale heterogeneous graph dataset, which could capture the comprehensive information from Wikidata.
2. The proposed method could  effectively learn the multi-attribute representation of constructed UniHG.
3. The proposed method significantly outperforms baselines.

---

> ### Author Rebuttal · Authors · 2025-07-30
>
> Thank you for your positive feedback regarding our work on the large-scale heterogeneous graph dataset, effective learning of multi-attribute representations, and significant performance improvements over baselines. For your concerns, we make the following responses.
>
> ### **Q**: My main concerns are the motivation behind this paper—why do we need to construct a heterogeneous graph? Is using a graph necessary for the same task? What are the advantages of structuring the data as a graph compared to the original text data? Additionally, the experimental results of this dataset on LLMs with text data should also be reported.
> ﻿
> **A**: Thanks for your valuable comments, we would like to clarify the motivation of this work by addressing your questions in detail:
>
> - Since entities in wikidata possess rich semantic relationships, a graph-based representation offers a more intuitive and structured way to model contextual information. And, it allows for efficient message propagation to capture long-distance associations. The diverse types of entities and edges naturally form a heterogeneous graph (please refer to Section 2.2).
>
> - In contrast to raw text, graphs provide a simpler and more explicit way to represent relationships, while enabling efficient long-range message propagation. Additionally, in practice, we also observe improvements in storage and computational efficiency: storage: Wikidata raw text (1.7 TB) vs. UniHG (37.5 GB); computation: prohibitively slow for text-based training even with 77M nodes (1.7T for raw text), whereas graph-based modeling remains efficient. Thus, for the same task involving complex associations, graph modeling should be more suitable than direct text learning.
>
> - Following your suggestion, we add the comparison with LLM on text data:
>
> i) 2,000 Labels Text Classification Task on UniHG-1M:
> |Methods|Acc|Recall|F1 Score|
> |---|---|---|---|
> |text encoder + linear classifier|61.28|65.31|61.25|
> |HGD(ours)|75.41|75.95|82.64|
>
>
> ii) Knowledge Transfer Experiment on Amazon-book:
> |Methods|Recall@20|NDCG@20|
> |---|---|---|
> |text encoder + linear classifier|+4.11%|+5.72%|
> |UniHG|+8.46%|+11.48%|
> These results further validate the effectiveness of our approach in handling structured knowledge and complex relational tasks.

---

> > ### Comment · Reviewer_faTf · 2025-08-06
> >
> > The response has solved my concerns, hence I would like to raise my score.

---

> > > ### Author Response · Authors · 2025-08-06
> > >
> > > Thanks for your positive feedback and confirmation that concerns have been solved. We appreciate your time and valuable comments again.

---

### Official Review · Reviewer_vPhg · 2025-07-02

**Rating:** 5
**Confidence:** 4

**Summary:**

This paper introduces UniHG, a large-scale, universal heterogeneous graph dataset derived from Wikidata, comprising over 77 million nodes, 564 million directed edges, and 2,082 relation types. To support scalable representation learning, the authors propose Heterogeneous Graph Decoupling (HGD), a framework featuring an Anisotropic Feature Propagation (AFP) module and a graph-free attention mechanism. UniHG aims to overcome key challenges in heterogeneous graph learning: lack of large-scale universal benchmarks, semantic misalignment, and inefficient information propagation. Experiments show that HGD significantly outperforms existing baselines (e.g., GAT, HGT, SIGN) in node classification tasks, and that universal knowledge learned from UniHG improves performance on downstream recommendation tasks through cross-domain knowledge transfer.

**Dataset Code Accessibility:**

Yes

**Ethical Considerations:**

No, there are no or only very minor ethics concerns

**Final Justification:**

Given the rebuttal, I will maintain my positive rating.

**Limitations Weaknesses:**

1. The dataset's quality relies heavily on pretrained models (CLIP, GPT-4). If these tools introduce noise or bias, the final representations may inherit them — a risk not discussed.

2. With over 2,000 labels, real-world datasets may face extreme imbalance or ambiguity in multi-label classification, yet the paper does not explore mitigation strategies or error analysis.

3. More experiments on more baselines are needed to further evaluate the dataset.

**Strengths Contributions:**

1. UniHG is the largest heterogeneous graph dataset to date and uniquely focuses on cross-domain knowledge representation, unlike prior datasets that are domain-specific or limited in scale.

2. HGD outperforms competitive baselines across different dataset scales (1M to 77M nodes), achieving up to 96.09% F1 score, and is 22.1x faster than HGT in inference speed.

3. Code and data are available, supporting further research and benchmarking in heterogeneous graph learning.

---

> ### Author Rebuttal · Authors · 2025-07-30
>
> Thanks for your positive appreciation about UniHG (largest dataset, higher performance, 22.1x faster inference speed, available code and data). Below, we address your concerns in detail.
>
> ### **Q1**: The dataset's quality relies heavily on pretrained models (CLIP, GPT-4). If these tools introduce noise or bias, the final representations may inherit them — a risk not discussed.
>
> **A1**: We sincerely thank you for this suggestion. As the universal SOTA models, CLIP and GPT-4 have been extensively validated for robustness and applied on various benchmark tasks. Further, we have taken specific strategies to enhance representation robustness:
>
> i)**Manual Checking**: The cluster descriptions generated by pretrained models were further verified manually, for example, per mapping (from sample to cluster) was checked by at least 3 persons (please see Section 2.4 in the manuscript). This human verification process can largely reduce potential noise/bias.
>
> ii)**Prompt Engineering**: Prompt templates could enforce structured outputs (please refer to the text reorganization module in Figure 1) , enhancing noise resilience.
>
> By incorporating manual verification and prompt design, we can reduce the risk of potential noise/bias propagation from foundation models. Following your suggestions, we make sure to add the above discussion in the next version.
>
>
> ### **Q2**: With over 2,000 labels, real-world datasets may face extreme imbalance or ambiguity in multi-label classification, yet the paper does not explore mitigation strategies or error analysis.
>
> **A2**: Thank you for your comment. The *original* labels with the quantity 74,666 indeed suffer from class imbalance or ambiguity, as you noted. To address this issue, in our work, we have performed the following operations:
>
> - **Semantic-similar merging**: During preprocessing, we consolidated semantically identical tags (e.g., "cat" and "kitten") and merged rare labels (occurrence < 0.1% of samples) into broader categories based on semantic similarity. After merging, the original label quantity was reduced to 2,000 labels. This largely alleviates long-tail effects while preserving semantic coherence. Please refer to some statements in lines 130-156.
>
> - **Focal Loss Usage**: During model training, we employed focal loss on the merged labels to further alleviate the long-tail distribution. Please refer to more details in our released code.
>
> Following your suggestion, we will re-clarify these points in the next version.
>
>
> ### **Q3**: More experiments on more baselines are needed to further evaluate the dataset.
>
> **A3**: Thanks for your comments. Here we add a comparison with RA-HGNN [Zhao et al., 2024]. The results on our dataset, UniHG, are as follows:
> ﻿Acc (1M) 55.60, Recall (1M) 55.35, F1 Score (1M) 58.17 Acc (10M) 56.60, Recall (10M) 58.99, F1 Score (10M) 51.75, Acc (Full) 54.87, Recall (Full) 56.64, F1 Score (Full) 58.80.
> As shown, the results are obviously lower than those achieved by our proposed method.
>
> Besides, we also investigate other recent works, e.g., BAB-GSL [Liu et al., 2025] and Revisiting Multi-view Learning [Zou et al., 2024], but find them impractical for very large-scale graph like UniHG. BAB-GSL relies on dense adjacency-matrix computations, which easily lead to out-of-memory errors. For UniHG (with over 77.3M nodes), this approach is prohibitively expensive. Revisiting Multi-view Learning is a meta-path-based method, which is infeasible for UniHG due to the combinatorial explosion of meta-paths. As discussed in Appendix §D.2, this would require about 74.95TB of memory, making it computationally intractable.
>
>
> We make sure to add these points in the revision.

---

> > ### Comment · Reviewer_vPhg · 2025-08-04
> >
> > Thanks for the rebuttal, I will maintain my rating on this paper.

---

> > > ### Author Response · Authors · 2025-08-04
> > >
> > > Thanks for your positive feedback. We appreciate your time and valuable comments again.

---

### Note · Authors · 2025-08-14

Dear ACs and Reviewers,

We would like to express our sincere gratitude to you for careful reading, constructive comments, valuable suggestions, and thoughtful evaluation on our submission.

In this work, we introduce UniHG, the largest universal heterogeneous graph dataset designed to date to advance heterogeneous graph representation learning and cross-domain knowledge mining. We further propose the Heterogeneous Graph Decoupling (HGD) framework to learn large-scale graph representations and enable effective knowledge transfer across eight downstream tasks. We believe our work makes enough contributions to the field of Large-scale Graph Representation Learning and aligns well with the standards of NeurIPS.

In the initial review phase, the strengths in innovation, theoretical contribution, dataset efforts and method effectiveness are appreciated by most reviewers. During the rebuttal and discussion phase, we thoroughly addressed all major concerns raised by the reviewers by providing additional experiments, analyses and clarifications that further consolidate this work. At the same time, all reviewers offered positive follow-up comments, explicitly acknowledging our rebuttal efforts and raising their scores or keeping the positive scores.

We sincerely appreciate all reviewers for reaching a positive consensus based on the rebuttal discussion. We ensure the revised manuscript incorporates the additional experimental results, discussion, and citations as suggested, further strengthening the validity and clarity of this paper.

Thank you once again for providing these valuable comments and suggestions!

Best Regards,

Authors

---

### Decision · Program_Chairs · 2025-09-18

**Decision:**

Accept (poster)

**Comment:**

The paper has been intensively discussed and finally all reviewers expressed support to accept. So an accept is recommended.